# Using performance art to promote intergroup prosociality by cultivating the belief that empathy is unlimited

Yossi Hasson [1,2,4] ✉, Einat Amir [3,4], Danit Sobol-Sarag[2], Maya Tamir[1] & Eran Halperin[1]

Empathy is important for resolving intergroup conflicts. However, people often tend to feel less empathy toward people who do not belong to their social group (i.e., outgroup members). We propose that this tendency is due, in part, to the belief that empathy is a limited resource. To overcome this issue, we develop an intervention synthesizing psychology and art to increase the belief that empathy is unlimited. In six studies ($n = 2118$), we find that the more people believe empathy is limited, the less outgroup empathy they experience. Moreover, leading people to believe that empathy is unlimited increase outgroup empathy, leads to greater support for prosocial actions toward outgroup members, and encourages more empathic behaviors toward outgroup members in face-to-face intergroup interactions. These intervention effects are observed across various intergroup contexts involving different ethnic, national, religious, and political groups. Thus, changing beliefs about empathy may improve intergroup relations, and conveying this belief through art may promote social change.

Intergroup tensions and conflicts involve a range of negative social outcomes, from stereotypes, prejudice, and discrimination, to deportation and violence. These outcomes, which are targeted toward individuals due to their affiliation with one group and not another, are often exacerbated by the lack of empathy toward outgroup members[1]. Empathy is defined as understanding (also known as cognitive empathy or perspective taking)[2] and sharing the feelings and thoughts of others (also known as affective or emotional empathy)[3], which often involves feelings of sympathy and compassion for others' suffering (also known as an empathic concern)[2,4]. For a full review of the definitions of empathy, see ref. 5. Outgroup empathy refers to empathy towards people who do not belong to one's social group[6] and plays an important role in fostering conflict resolution and promoting more positive intergroup relations[7]. Despite its benefits in promoting prosocial behavior between groups[8], empathy is limited by group boundaries[9]. People often feel less empathy for individuals whom they

perceive as being outside their group (i.e., outgroup members), compared to individuals whom they perceive as being in their group (i.e., ingroup members). This empathy bias exists in many intergroup contexts in which people empathize less with those who differ in their ethnicity, political ideology, religion, skin color, gender, and other social categories[10,11]. The bias may increase in violent and conflictual intergroup contexts, in which some people may experience counter-empathic feelings, such as pleasure in response to outgroup suffering[12]. Understanding the psychological mechanism that drives this bias can help in finding ways to attenuate the intergroup empathy bias and increase outgroup empathy.

We propose that people feel less empathy toward outgroup compared to ingroup members, in part, because they consider empathy as a limited resource. Resources are typically considered as something concrete and tangible, but the term resource can also be used in relation to psychological faculties. For instance, according to

[1]Psychology Department, The Hebrew University, Mount Scopus Jerusalem 9190501, Israel. [2]School of Psychology, Reichman University, Herzliya 4610101, Israel. [3]Department of Art and Media & Department of Neuroscience and Biomedical Engineering, Aalto University, FI-00076 Espoo, Finland. [4]These authors contributed equally: Yossi Hasson, Einat Amir ✉e-mail: yossi.hasson@mail.huji.ac.il

the resource depletion theory, resources underlying self-regulation are limited and using them for one task leaves less available resources for others[13]. In contrast, there are studies showing that effortful self-regulation is not necessarily limited[14,15]. Such studies show that the exertion of self-regulation does not invariably reduce subsequent self-regulation[16] and sometimes can even increase it[17]. For instance, when people are motivated by incentives to control themselves, they do not show depletion effects compared to people who are not incentivized[18].

Regardless of the actual limitation of resources, subjective beliefs about their capacity can have behavioral implications. Believing a resource is limited can lead people to use it sparingly and selectively, while believing a resource is unlimited can lead people to use it generously and unselectively[19,20]. For example, people who believed the capacity for self-control is unlimited did not show diminished self-control after a depleting effortful task[21]. Moreover, leading people to believe that willpower is unlimited led to increased self-regulation, as demonstrated in eating behaviors and academic tasks.

In terms of empathy and other prosocial emotions, a common assumption is that the capacity to feel them is limited[22–24]. Previous studies imply that empathy is limited because it requires cognitive and mental resources that can be depleted over time[25], and being too empathic toward one person can reduce the capacity for bearing the suffering of others[26]. Indeed, some indirect evidence suggests that there may be trade-offs associated with empathy at work and at home. Workers who take time to listen to coworkers' problems and help others with heavy workloads feel less capable of connecting with their families due to emotional exhaustion[27].

Whether empathy is objectively limited or not, how much empathy people experience may also depend on their subjective beliefs about empathy. For example, previous research showed that when people believe empathy can be developed (i.e., people with a growth mindset), they are more likely to empathize with others[28,29]. According to this theory, people with a growth mindset (as opposed to people with a fixed mindset) are typically more motivated by learning goals and may thus be more likely to regard the challenging intergroup context as an opportunity for growth. While this theory was found to be more relevant in interpersonal relations[29], our theory focuses on empathy in intergroup relations. Specifically, we propose a unique mechanism in which people who believe empathy is a limited resource are less likely to feel empathy toward outgroup members because they think it would come at the expense of their own ingroup members.

People tend to believe resources are limited even when they are not. This bias often occurs in regard to desirable resources[30], such as empathy. People tend to value empathy and its related qualities, like kindness and generosity, as desirable traits that are favored and appreciated by society[31]. Thus, it is likely that people are biased in their beliefs regarding the limited capacity for empathy. When resources are believed to be limited and scarce, it increases a zero-sum mindset in which gains for one are perceived as losses for the other[30]. At the intergroup level, gains for one group are perceived as losses for the other group. For example, when resources are perceived to be limited during a health or financial crisis, such as during the COVID-19 crisis[32], some people may tend to discriminate more against outgroup members because they are seen as potential competitors who might deplete resources[33]. Even when resources are not scarce, people whose world views are characterized by zero-sum thinking (e.g., high in social dominance orientation) express low levels of empathy toward outgroup members[34,35].

Existing interventions that aim to increase empathy often use explicit instructions to empathize with a specific person or group[36]. However, such interventions can sometimes backfire, leading to resistance and self-serving behavior[37]. The negative effect of such direct interventions often occurs in intergroup conflictual contexts[38] and competitive interactions[39], when there is a greater threat due to

perceived dissimilarities with the person or group in need[40], and among those who are highly identified with their own group[41].

Building on these ideas, we hypothesized that an indirect approach that does not mention any specific intergroup context would be more powerful in increasing outgroup empathy. Our hypothesis was that the more people believe empathy is a limited resource in general, the more they think that empathy toward outgroup members comes at the expense of their own group members. Consequently, such individuals would be less likely to empathize with outgroup members and more likely to empathize with ingroup members. We further hypothesized that a psychological intervention fostering the belief that empathy is an unlimited resource would promote empathy and prosociality toward outgroup members and attenuate the intergroup empathy bias in real-world conflict contexts.

When it comes to psychological interventions, it is particularly important to examine their effectiveness not only in the laboratory but also in field studies, due to the fact that lab studies do not necessarily generalize to real-life settings[42]. One way to do so is by implementing psychological interventions within ordinary events in which people attend and take active roles such as in performance art. Performance art is an art form created through live actions performed by the artist or other performers. The term "performance art" is used in its current meaning since the 1970s to describe a non-traditional art form that incorporates action and body movement as an alternative to the static nature of painting and sculpture[43]. By playing an active role, participants can engage with live art and re-imagine and restage the social rules, codes, and conventions regarding different social and political issues[44].

In this work, that includes six studies ($n = 2118$), we test whether the belief that empathy is an unlimited resource is associated with more empathic sentiments (Pilot Study) and more empathic reactions toward ingroup and outgroup members in need (Study 1). Next, we test whether leading people to believe that empathy is unlimited attenuates the intergroup empathy bias and increased empathy toward an outgroup member (Studies 3 and 5) or multiple outgroup members in need (Studies 2 and 4). To test the generalizability of the findings, we examine distinct intergroup contexts: liberals and conservatives in the United States, social groups in Israeli society (i.e., Jewish secular, religious Jewish, Ultra-Orthodox, and Arab), and Syrian refugees who fled to the United States and US citizens.

Finally, moving beyond the traditional correlational and experimental designs that are used in Studies 1–3, we test our hypotheses in a real-world setting by conducting two large-scale field studies in the United States and in Israel. For these studies, a joint team of psychologists and artists, has developed an interdisciplinary concept that is a hybrid of psychological experiments and performance art in which the audience also consensually acts as the participants of the experiment. This concept enables us to test our hypotheses in a real-world setting by bringing people together from different social groups with the intention that they can also benefit from the psychological insights proposed in this project.

Our findings suggest that the belief about empathy as an unlimited resource is positively associated with greater empathy and prosocial behavioral tendencies toward outgroup members, and, consequently, with a decrease in intergroup empathy bias. This is true in various intergroup contexts, including different ethnic, national, religious, and political groups.

## Results
### Pilot Study
The goal of the pilot study was to provide support for our hypothesis that the belief that empathy is an unlimited resource is positively associated with empathy toward ingroup and outgroup members. As an initial examination, we assessed the association between the extent to which people believe empathy is unlimited and the level of empathic

**Table 1 | Simple correlations between belief about empathy as an unlimited resource and empathic sentiments toward ingroup and outgroups**

| Target groups toward whom empathic sentiments were measured | Jewish Secular Sample (n = 322) | Jewish Religious Sample (n = 509) | Arab Sample (n = 477) |
|---|---|---|---|
| Secular Jews | r = 0.104<br>95% CI [−0.01, 0.21]<br>p = 0.063, d = 0.21 | r = 0.202<br>95% CI [0.12, 0.28]<br>p < 0.001, d = 0.41 | r = 0.100<br>95% CI [0.01, 0.19]<br>p = 0.030, d = 0.2 |
| Religious Jews | r = 0.169<br>95% CI [0.06, 0.27]<br>p = 0.002, d = 0.34 | r = 0.224<br>95% CI [0.14, 0.3]<br>p < 0.001, d = 0.46 | r = 0.164<br>95% CI [0.08, 0.25]<br>p < 0.001, d = 0.33 |
| Ultra-Orthodox Jews | r = 0.174<br>95% CI [0.07, 0.28]<br>p = 0.002, d = 0.35 | r = 0.227<br>95% CI [0.14, 0.31]<br>p < 0.001, d = 0.47 | r = 0.152<br>95% CI [0.06, 0.24]<br>p = 0.001, d = 0.31 |
| Arabs | r = 0.183<br>95% CI [0.07, 0.28]<br>p = 0.001, d = 0.37 | r = 0.109<br>95% CI [0.02, 0.19]<br>p = 0.014, d = 0.22 | r = 0.049<br>95% CI [−0.04, 0.14]<br>p = 0.290, d = 0.1 |
| All outgroups | r = 0.224<br>95% CI [0.12, 0.33]<br>p < 0.001, d = 0.46 | r = 0.253<br>95% CI [0.17, 0.33]<br>p < 0.001, d = 0.52 | r = 0.163<br>95% CI [0.74, 0.25]<br>p < 0.001, d = 0.33 |
| All groups | r = 0.229<br>95% CI [0.12, 0.33]<br>p < 0.001, d = 0.47 | r = 0.281<br>95% CI [0.2, 0.36]<br>p < 0.001, d = 0.59 | r = 0.169<br>95% CI [0.08, 0.25]<br>p < 0.001, d = 0.34 |

The Pearson correlation coefficients reflect the relation between belief about empathy as an unlimited resource and empathic sentiments toward ingroup (highlighted in gray) and outgroups. All tests are two-tailed and corrected for multiple comparisons. Source data are provided as a Source Data file.

sentiments toward the main social groups dividing Israeli society (namely–Jewish Secular, Religious Jewish, Ultra-Orthodox, and Arab), which reflect the extent to which they generally experience empathic feelings toward these groups, unrelated to specific events or actions of the groups. We collected the data from a nationwide educational project that included a large sample of 1308 Israeli teenagers who self-identified as either Jewish Secular, Religious Jewish, or Arab. See Methods and SI Appendix for a full description of samples, power estimations, and measures of all studies. A series of simple bivariate correlations (with Bonferroni correction) revealed that the more participants believed empathy is unlimited, the more empathic sentiments they tended to feel toward members of any outgroup. No significant correlations were found between the belief that empathy is unlimited and empathy toward the ingroup, except among participants who self-identified as religious Jews (see Table 1).

The results indicate a moderate positive association between the belief in empathy as an unlimited resource and empathic sentiments toward outgroup members, whereas the association with empathic sentiments toward ingroup members was less consistent. Given these results regarding the association between general empathy sentiments and the belief that empathy is unlimited, we wanted to examine the link between the belief that empathy is unlimited and empathic reactions toward ingroup and outgroup members in need.

## Study 1

In light of the increasing political polarization in the United States[45] and the low level of outgroup empathy between the rival political parties[46], we tested whether the belief about empathy as an unlimited resource is associated with empathy toward liberals and conservatives who were injured during a political protest. We recruited 182 US participants, of whom 52.7% self-identified as liberal and 47.3% self-identified as conservative. In the first assessment, participants indicated their political ideology and their belief about empathy as an unlimited resource. In the second assessment, participants were randomly assigned to two conditions in which they read a fictional newspaper article about either political ingroup or outgroup members who were injured during a protest. After reading the article, participants rated their empathic reactions (i.e., empathy, sympathy, and compassion; from 1 = not at all, to 7 = very much) toward the injured people. We combined empathy, sympathy, and compassion to obtain a

reliable index of prosocial emotions that are often intertwined and involve caring for others[47]. However, it should be noted that these emotions are not identical and have some distinct properties[48,49] that can lead to different emotional and behavioral outcomes[50].

To examine the effect of the condition (ingroup vs. outgroup targets) and the moderating effect of belief about empathy as an unlimited resource on empathic reactions, we used Hayes's[51] PROCESS model 1. Participants' empathic reactions were significantly predicted by condition (b = −0.57, SE = 0.22, t(178) = −2.59, p = 0.01; 95% confidence interval [CI] = [−1.0, −1.36]), such that participants felt more empathy toward ingroup members compared with outgroup members (M = 5.10 vs. M = 4.60, respectively). Meaning, participants who self-identified as liberal felt more empathy toward liberals that were injured as compared to the conservatives that were injured, and participants who self-identified as conservative felt more empathy toward conservatives that were injured as compared to the liberals that were injured. More importantly, while there was no significant main effect of belief about empathy (b = 0.04, SE = 0.12, t(178) = 0.32, p = 0.75; 95% CI = [−0.2, 0.28]), there was a significant Condition × Belief interaction (b = 0.41, SE = 0.17, t(178) = 2.40, p = 0.017; 95% CI = [0.074, 0.76]; Fig. 1). Participants who believed empathy is more limited exhibited a stronger intergroup empathy bias, as indicated by more empathy toward ingroup members than toward outgroup members (t(178) = −3.52, p < 0.001; 95% CI = [−1.6, −0.45]). In contrast, no significant difference was found in empathy towards ingroup and outgroup members among participants who believed empathy was less limited (t(178) = 0.03, p = 0.98; 95% CI = [−0.64, 0.65]).

After demonstrating a significant association between the belief about empathy as unlimited and empathic reactions toward outgroup members, in Study 2, we tested the potential causal effect of such beliefs on empathic reactions toward outgroup members in need.

## Study 2

This study tested whether promoting the belief about empathy as an unlimited resource can causally shape empathic reactions toward multiple cases of outgroup members in need. To increase our confidence in the generalizability of the findings so far, we conducted this study in a different intergroup context. We tested whether changing the belief about empathy as an unlimited resource among US participants would increase empathy toward Syrian refugees who fled to the United States after being tortured in their homeland.

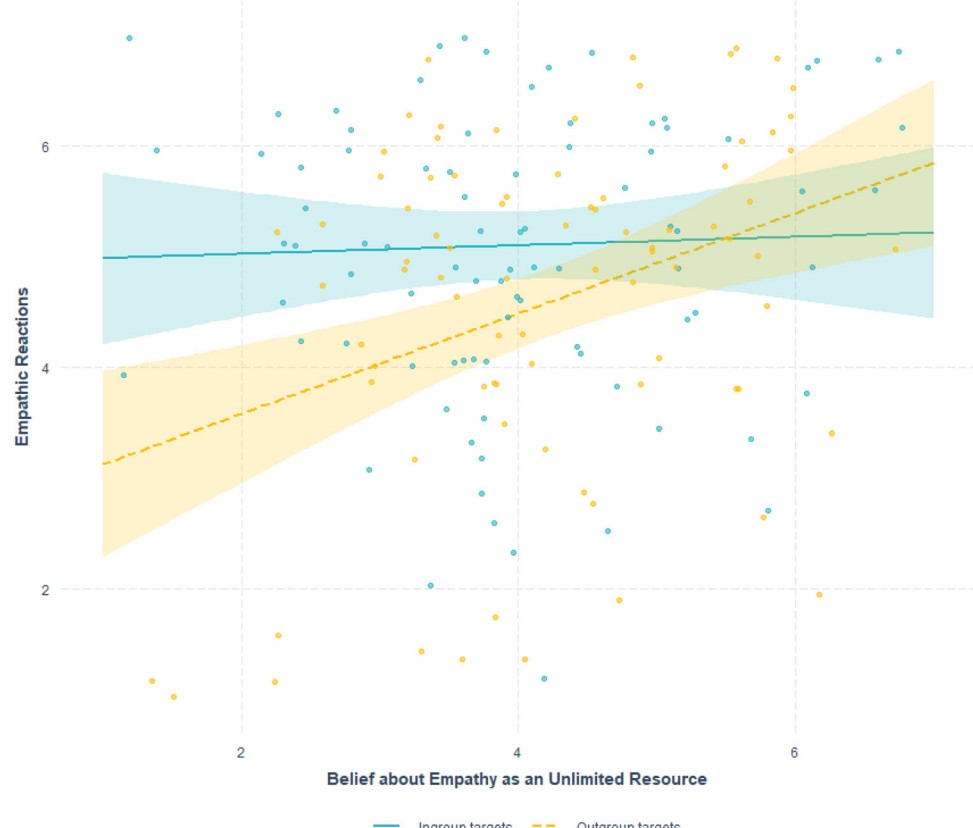

**Fig. 1 | The interactive effect of condition (ingroup targets, *n* = 92 vs. outgroup targets, *n* = 90) and belief about empathy as an unlimited resource on participants' empathic reactions toward ingroup and outgroup members.** A significant Condition × Belief interaction (*b* = 0.41, SE = 0.17, *t*(178) = 2.40, *p* = 0.017; 95% CI = [0.074, 0.76]). A significant difference between empathy towards ingroup and outgroup members among participants who believed empathy is more limited

(*t*(178) = −3.52, *p* < 0.001; 95% CI = [−1.6, −0.45]). No significant difference in empathy towards ingroup and outgroup members among participants who believed empathy was less limited (*t*(178) = 0.03, *p* = 0.98; 95% CI = [−0.64, 0.65]). Error bands represent a 95% confidence interval. All tests are two-tailed. Source data are provided as a Source Data file.

Two hundred US participants were randomly assigned to one of two conditions, limited or unlimited empathy, in which their belief about the limit of empathy was manipulated. Following the manipulation, participants read four empathy-inducing testimonies, based on real stories of Syrian refugees who were tortured in their homeland (see SI Appendix for the manipulation and scenarios). After reading each testimony, participants rated their empathic reactions toward the Syrian refugee (on a scale of 1 = not at all, to 7 = very much). Finally, participants completed a manipulation check using the scale of belief about empathy as an unlimited resource used in Study 1.

Regarding the manipulation check, an independent-samples t-test revealed a significant effect of condition, *t*(1, 198) = −4.11, *p* < 0.001, *d* = 0.58, such that participants in the unlimited condition believed empathy is unlimited (*M* = 4.92, SD = 1.89) more than did those in the limited condition (*M* = 3.9, SD = 1.58).

To test whether the manipulation influenced empathic reactions in response to each Syrian refugee's testimony, we ran a repeated-measures ANOVA with testimony order (1–4) as a within-participant variable, and the condition (unlimited vs. limited) as a between-participants variable. We used testimony order as the within-participant variable because all testimonies were presented in a counterbalanced order to rule out the possibility that the content of the testimonies influences the results. We found a significant main effect of condition on empathic reactions, *F*(1, 198) = 8.93, *p* = 0.003, *d* = 0.423. On average, participants in the unlimited condition felt more empathy toward the outgroup members (*M* = 5.82, SD = 1.43), compared to those in the limited condition (*M* = 5.18, SD = 1.60). Pairwise comparisons between the effects of limited and unlimited conditions

on empathy in each testimony were all significant (Testimony #1: *p* < 0.001; Testimony #2: *p* = 0.003; Testimony #3: *p* = 0.037; Testimony #4: *p* = 0.013). Moreover, we found a significant Testimony Order × Condition interaction, *F*(1, 196) = 2.78, *p* = 0.042, *d* = 0.41 (Fig. 2). While empathy changed across stories in the limited condition (between testimonies #1 and #4; *p* = 0.021), empathy remained stable in the unlimited condition, and there were no significant differences across stories (between testimonies #1 and #4; *p* = 0.917).

Study 2 showed that promoting the belief about empathy as an unlimited resource led to greater outgroup empathy compared to the belief that empathy is limited. In Study 3, we tested the causal effect of such beliefs on empathic reactions toward both ingroup and outgroup members in need. Given the relatively high level of outgroup empathy found in Study 2 (maybe due to the severity of the testimonials or because the intergroup context is less conflictual), we wanted to examine the effect of the intervention in a more challenging context where empathy is expected to be lower. We also moved beyond empathic reactions per se and tested the effect of the manipulation on prosocial support. In addition, we included a natural condition in which we provided no information about the limited or unlimited nature of empathy.

## Study 3

This study tested whether promoting the belief about empathy as an unlimited resource can increase empathic reactions and support for prosocial actions. One hundred and fifty participants who self-identified as Israeli Jews were randomly assigned to one of three conditions: (a) The belief that empathy is limited, (b) the belief that

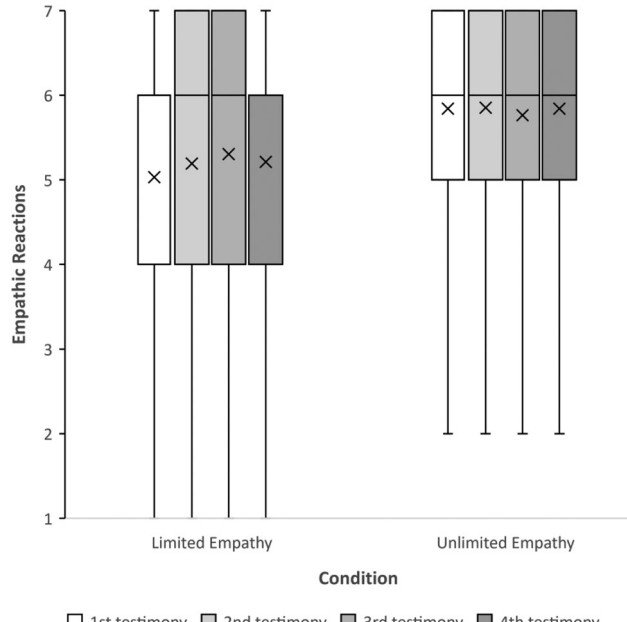

**Fig. 2 | Empathic reactions toward outgroup targets, as a function of Condition (i.e., limited empathy, *n* = 99; and unlimited empathy, *n* = 101) and order of empathy-inducing testimonies (i.e., 1–4).** A significant main effect of condition on empathic reactions, $F(1, 198) = 8.93$, $p = 0.003$, $d = 0.423$ (a repeated-measures ANOVA). The plot presents the maximum and minimum values as whiskers, the interquartile range as the vertical length of each box, the mean as the x marker, and the median as the horizontal line within each box. All tests are two-tailed. Source data are provided as a Source Data file.

empathy is unlimited, or (c) a control condition, where the belief about empathy was not manipulated. Following the manipulation, participants read two fictional empathy-inducing articles about a family in need (see SI Appendix for the manipulation and scenarios). One of the articles described a Jewish family (i.e., ingroup) and the other article described an Arab family (i.e., outgroup). After reading each article, participants rated their empathic reactions (i.e., empathy, sympathy, and compassion; from 1 = not at all, to 7 = very much), and support for prosocial actions (e.g., "Support the authorities provide help to this family"; "Contacting the relevant Minister to handle such cases"). Finally, participants completed a manipulation check, using the scale of belief about empathy as an unlimited resource used in Study 1.

Regarding the manipulation check, a one-way ANOVA revealed a significant main effect of the condition, $F(2, 147) = 29.60$, $p < 0.001$, $d = 1.27$. Post-hoc analyses indicated a significant difference between the three conditions, such that compared to the control condition ($M = 3.52$, SD = 1.25), participants in the unlimited empathy condition agreed more with statements that empathy is an unlimited resource ($p = 0.008$, $M = 4.42$, SD = 1.56), and participants in the limited empathy condition agreed less that empathy is unlimited ($p < 0.001$, $M = 2.27$, SD = 1.43).

To test whether the manipulation influenced empathic reactions, a repeated-measures ANOVA with the target identity (ingroup vs. outgroup) as a within-participant variable, and condition (limited, unlimited, and control) as a between-participants variable revealed a significant Target identity × Condition interaction, $F(2, 147) = 4.50$, $p = 0.013$, $d = 0.5$ (Fig. 3a). Participants felt more empathy toward ingroup members and less empathy toward outgroup members in the limited condition ($M = 4.21$, SD = 1.10; $M = 3.52$, SD = 1.20; $p < 0.001$) and in the control condition ($M = 4.31$, SD = 1.12; $M = 3.57$, SD = 1.49; $p < 0.001$). In contrast, no significant difference was found in empathy towards ingroup and outgroup members among participants in the

unlimited condition ($M = 4.16$, SD = 1.2; $M = 4.08$, SD = 1.15; $p = 0.654$). Importantly, the intergroup empathy bias in the unlimited condition attenuated as a result of an increase in outgroup empathy (unlimited vs. control, $p = 0.047$; unlimited vs. limited, $p = 0.028$) and not a decrease in ingroup empathy (unlimited vs. control, $p = 0.53$; unlimited vs. limited, $p = 0.83$).

To test whether the manipulation also influenced support for prosocial actions, a repeated-measures ANOVA with the target identity (ingroup vs. outgroup) as a within-participant variable, and the condition (limited, unlimited, and control) as a between-participants variable revealed a significant Target identity × Condition interaction, $F(2, 147) = 4.76$, $p = 0.01$, $d = 0.51$ (Fig. 3b). Participants were more likely to support prosocial actions toward ingroup members than toward outgroup members in the limited condition ($M = 4.34$, SD = 1.32; $M = 3.72$, SD = 1.49; $p < 0.001$) and in the control condition ($M = 4.23$, SD = 1.28; $M = 3.66$, SD = 1.51; $p = 0.002$). In contrast, no significant difference was found in support for prosocial actions toward ingroup members and outgroup members among participants in the unlimited condition ($M = 4.14$, SD = 1.06; $M = 4.21$, SD = 1.15; $p = 0.72$). In addition, we found a marginally significant increase in support for the outgroup (unlimited vs. control, $p = 0.052$; unlimited vs. limited, $p = 0.075$) and no significant difference in support for the ingroup (unlimited vs. control, $p = 0.72$; unlimited vs. limited, $p = 0.42$).

To examine whether the effect of beliefs about empathy as an unlimited resource on the bias in support for prosocial actions was mediated by the intergroup empathy bias, we conducted a mediation analysis employing the procedure of Hayes[51] PROCESS model 4. The model was specified with a condition as the independent variable, intergroup empathy bias (ingroup empathy minus outgroup empathy) as the mediator variable, and intergroup bias in support for prosocial actions (ingroup support minus outgroup support) as the outcome variable (Fig. 4). The total effect of condition on the intergroup bias in support for prosocial action ($b = -0.34$, 95% CI = [−0.59, −0.1], $t(148) = -2.7$, $p = 0.006$) was reduced when intergroup empathy bias was added as a mediator ($b = -0.19$, 95% CI = [−0.41, 0.03], $t(147) = -1.7$, $p = 0.085$). The indirect effect through the mediator was statistically different from zero ($b = -0.15$, 95% CI = [−0.2, −0.04]). This means that among participants in the unlimited condition, there was less intergroup empathy bias, which in turn, was associated with less intergroup bias in support of prosocial actions.

Study 3 demonstrated the positive effects of beliefs about empathy as an unlimited resource on empathic reactions and support for prosocial actions toward the outgroup, without influencing the support of the ingroup. Consequently, the manipulation in the unlimited condition led to the attenuation of the intergroup empathy bias. The intergroup empathy bias found in the control condition was similar to the bias found in the limited condition, suggesting that, by default, people believe that empathy is more limited. Moreover, since the beliefs about empathy affect only empathic reactions toward outgroup, but not toward ingroup members, the concern about demand characteristics of our manipulation is ruled out. Given the potential of such findings for improving intergroup relations in the real world, we proceeded to test our manipulation outside the laboratory. We were interested in developing a practical method that might effectively convey the idea that empathy is unlimited to the general public.

## Studies 4–5

In these studies, we tested our hypotheses using a unique method of field experiments that synthesize psychology and art. We developed a hybrid concept of psychological experiments that are embedded in performance art. Specifically, we chose a participatory form of performance art, meaning that the audience is invited to take part and become active in work, such that their experience and responses are incorporated in the final artistic result[52].

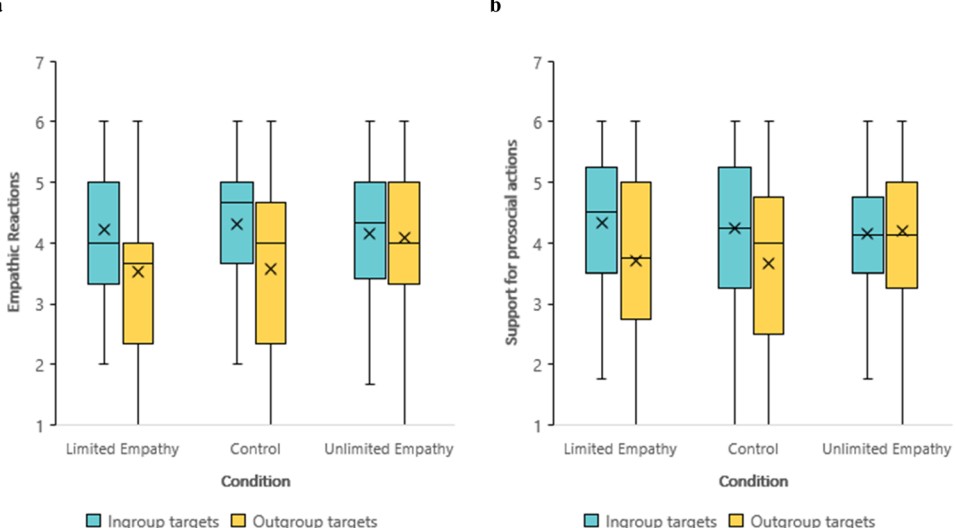

**Fig. 3 | Empathic reactions and support for prosocial actions. a** Empathic reactions toward ingroup and outgroup targets, as a function of condition (i.e., limited empathy, $n = 51$; unlimited empathy, $n = 52$; and control, $n = 47$). A significant target identity × condition interaction, $F(2, 147) = 4.50$, $p = 0.013$, $d = 0.5$. A significant difference in empathy toward ingroup and outgroup members in the limited condition ($p < 0.001$) and in the control condition ($p < 0.001$). No significant difference in empathy towards ingroup and outgroup members among participants in the unlimited condition ($p = 0.654$). **b** Support for prosocial actions toward ingroup and outgroup targets as a function of condition (i.e., limited empathy, $n = 51$; unlimited empathy, $n = 52$; and control, $n = 47$). A significant target

identity × condition interaction, $F(2, 147) = 4.76$, $p = 0.01$, $d = 0.51$. A significant difference in support for prosocial actions toward ingroup and outgroup members in the limited condition ($p < 0.001$) and in the control condition ($p = 0.002$). No significant difference in support for prosocial actions toward ingroup and outgroup members in the unlimited condition ($p = 0.72$). The plots present the maximum and minimum values as whiskers, the interquartile range as the vertical length of each box, the mean as the x marker, and the median as the horizontal line within each box. All tests are two-tailed. Source data are provided as a Source Data file.

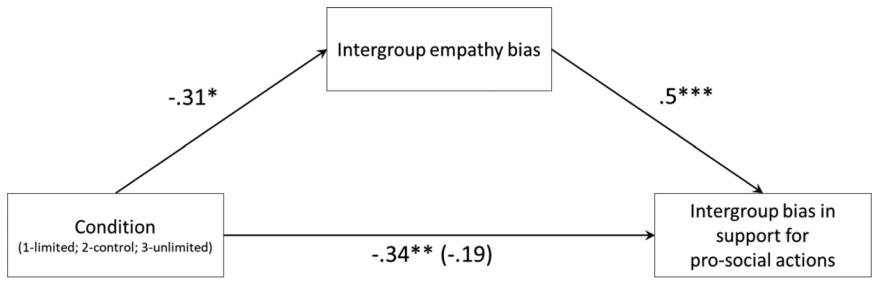

**Fig. 4 | A mediation model of the effect of condition on intergroup bias in support for pro-social actions through intergroup empathy bias.** A mediation analysis shows that intergroup empathy bias mediates the effect of manipulated

beliefs about empathy as a limited resource on intergroup bias in support of prosocial actions. * indicates $p < 0.05$, **$p < 0.01$, ***$p < 0.001$. All tests are two-tailed. Source data are provided as a Source Data file.

The concept of performance-experiment was developed for four main reasons. First, performance art provides an opportunity to bring together people from different social groups and generate intergroup interactions that are required for testing our manipulation outside the laboratory. Second, the active participation of the audience in the performance-experiment provides a platform for sharing new findings in science and educating the public about empathy. Indeed, throughout these performance-experiment projects, through the creation of creative, engaging debriefings (using actors and visual aids) and through sharing the experiments' results with the participants, we make the audience part of an educational process that could potentially contribute to social change. Third, we wanted to examine the effectiveness of the hybrid platform in implementing a real-world psychological intervention, such as the belief about empathy as an unlimited resource, and pave the way for future interventions. Lastly, to make our psychological intervention suitable and sustainable for use in the real-world, we wanted to have a platform that is appealing enough for people outside the lab setting,

when there is no external, monetary incentive, to undergo the intervention.

Studies 4 and 5 were conducted by joint teams of psychologists and artists using the concept of performance-experiment. Study 4 tested the effect of the belief about empathy as an unlimited resource on outgroup empathy in the context of national groups (US citizens and Syrians), whereas Study 5 was conducted in the context of ethnic groups (Jews and Arabs).

## Study 4
In this study, we sought to replicate the findings of Study 2 in a more realistic setting as part of a performance art that was also planned and designed as a psychological experiment. Unlike conventional studies, here we trained actors (instead of research assistants) to perform an empathy-inducing scene uniformly and simultaneously. It was also important for us to use real stories (and not fictional ones as often used in lab studies), so that the audience could later read more about them and about ways to help the outgroup members. In addition, the

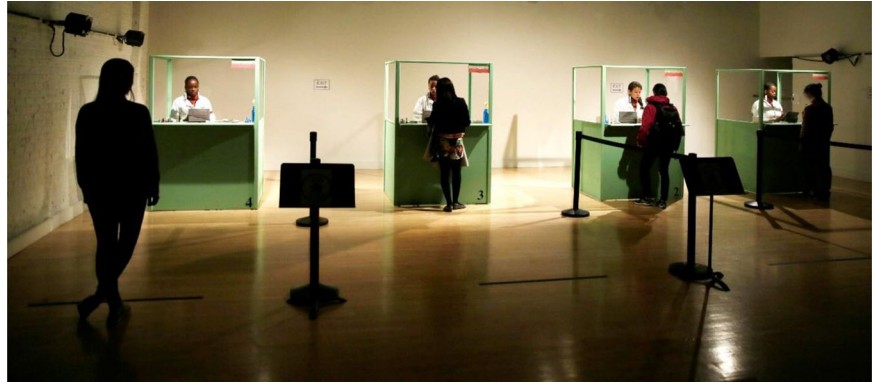

**Fig. 5 | The performance-experiment setting in Study 4.** Participants met actresses who were seated in a theatrical set inspired by the architecture of "airport immigration booths" welcoming one participant at a time for an "entry" interview.

debriefing was used to explain more than the study itself, but also the main findings thus far, and the hybrid concept of art and psychology and its benefits.

We invited potential audience members to participate in an artistic event which is also a psychological experiment. Upon arrival at a performance venue in Chicago, each participant was welcomed individually by our staff and were given a consent form to sign. The 108 US participants were randomly assigned to one of two conditions —limited empathy or unlimited empathy. Participants met actresses who were seated in a theatrical set inspired by the architecture of "airport immigration booths" welcoming one participant at a time for an "entry" interview (see Fig. 5). The interview included several questions that led participants to believe that empathy is limited or unlimited (similar to Study 2. A video documentation with the experimental manipulation is available at https://bit.ly/As-Much-As-You-Want). Following the manipulation, participants were presented with four empathy-inducing testimonies based on real stories of Syrian refugees who were tortured in their homeland (similar to Study 2 with an addition of a portrait of each refugee). After each testimony, participants rated their empathic reactions (same as in Study 2). Next, participants completed a manipulation check, using the scale of belief about empathy as an unlimited resource used in Study 1. Following the manipulation check, the actresses engaged in further dialog with each participant, following a theatrical script that was designed to enrich the artistic experience. Finally, participants received an envelope with more information about the refugees' testimonies and ways in which they can help refugees (i.e., https://www.rescue.org/how-to-help). Upon completion, participants were debriefed and had an open discussion with the research team (both psychologists and artists) that explained the concept of outgroup empathy, our findings thus far, and the idea of combining art and psychology.

Regarding the manipulation check, an independent-samples $t$-test revealed a significant effect of condition, $t(1, 106) = -2.35$, $p = 0.02$, $d = 0.45$, such that participants in the unlimited condition believed empathy is unlimited ($M = 4.12$, SD = 1.51) more than participants in the limited condition did ($M = 3.44$, SD = 1.49).

To test whether the manipulation influenced empathic reactions, a repeated-measures ANOVA with testimony order (1–4) as a within-participant variable, and condition (unlimited vs. limited) as a between-participants variable revealed a significant main effect of condition on empathic reactions, $F(1, 106) = 6.34$, $p = 0.013$, $d = 0.49$. On average, participants in the unlimited condition felt more empathy toward the refugees ($M = 6.08$, SD = 1.13), compared to participants in the limited condition ($M = 5.5$, SD = 1.25). Moreover, we found a significant Testimony order × Condition interaction, $F(1, 106) = 3.97$, $p = 0.008$, $d = 0.39$ (Fig. 6). While changed across stories in the limited

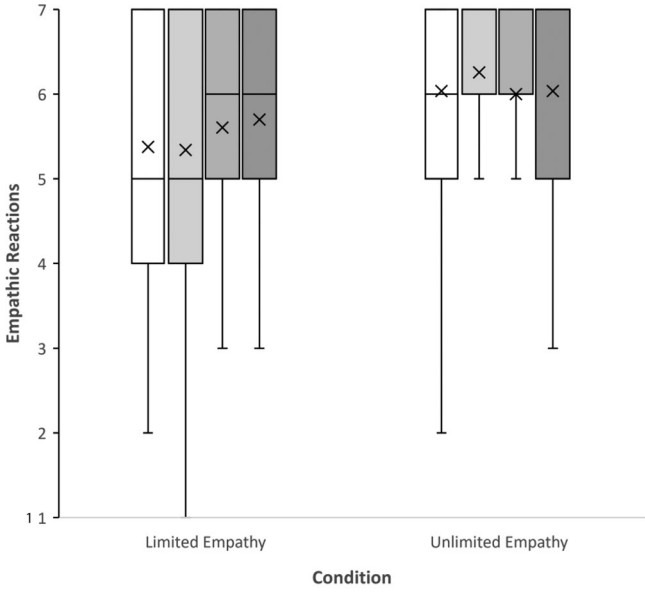

**Fig. 6 | Empathic reactions toward outgroup targets, as a function of condition (i.e., limited empathy, $n = 53$; and unlimited empathy, $n = 55$) and order of empathy-inducing testimonies (i.e., 1–4).** A significant main effect of the condition on empathic reactions, $F(1, 106) = 6.34$, $p = 0.013$, $d = 0.49$. The plot presents the maximum and minimum values as whiskers, the interquartile range as the vertical length of each box, the mean as the x marker, and the median as the horizontal line within each box. All tests are two-tailed. Source data are provided as a Source Data file.

condition (between testimonies #1 and #4; $p = 0.021$), empathic reactions in the unlimited condition remained stable, and there were no significant differences across stories (between testimonies #1 and #4; $p = 1$).

Study 4 replicated the results in Study 2 in a more realistic context by showing how the belief about empathy as unlimited led US participants to experience more empathy toward outgroup members (i.e., Syrian refugees) to whom they might feel less empathy. Moreover, the performance-experiment was our first attempt to communicate the findings about the effect of the belief about empathy and its impact on outgroup empathy to the public. Next, we conducted Study 5 to examine the causal effect of such beliefs about empathy on empathic reactions and behavior toward both ingroup and outgroup members in need during face-to-face intergroup interactions among a larger audience.

## Study 5

We conducted the second performance art-experiment as part of an art festival in Jerusalem called "MEKUDESHET" (i.e., "SACRED"). Jerusalem is an ethnically diverse, multicultural city. These characteristics render Jerusalem an ideal location to test whether the belief about empathy as an unlimited resource can increase empathy toward outgroup members and attenuate the intergroup empathy bias. The experiment was conducted as participants were in direct contact with ingroup (i.e., Jews) and outgroup (i.e., Arabs) members in need. Among the people who registered and attended the event (-700), we randomly selected 176 participants who self-identified as Israeli Jews to take an active part in the performance-experiment. The other attendants observed the performance-experiment as audience members.

The performance-experiment was planned 10 months in advance. During this time, we determined how to embed psychological manipulation into the artistic experience, how to deliver the main findings to the public in an engaging and creative form, and trained actors who self-identified as Jews or Arabs on how to perform empathy-inducing scenes uniformly and simultaneously. While this event was expected to be highly engaging, we designed the scenes and the instructions in a way that would focus on the other person's situation rather than leading participants to imagine themselves in the same situation. This was an important aspect because using "imagine-self" instructions are known to increase personal distress, which is negatively linked to empathic concern and prosocial behavior[53]. The event included four main phases. First, in the introduction phase, part of the audience was randomly selected to actively participate in the performance-experiment while the others observed it as spectators. Those who actively participated in the performance-experiment were randomly assigned to one of two conditions that were conveyed in the form of actresses' monologues in a theatrical set: an experimental condition in which the belief about empathy as an unlimited resource was manipulated, or a control condition that only included an explanation about empathy. Second, in the intergroup encounter phase, after the manipulation (or control), each participant entered a cabin in which they met separately with an Arab actor and with a Jewish actor, each of whom shared a sad personal story that was written for the purpose of the performance (see SI Appendix for manipulation and scenarios). After each story, participants were asked if they wanted to end the encounter with no interpersonal touch, with a handshake, or with a hug. Then, participants rated their empathic reactions (same as in Study 3). Third, in the science outreach and debriefing phase, all attendees (-700) watched a video art produced for this event that reviewed the literature on outgroup empathy, performed by actresses and dancers, and a live stage performance sharing the main findings of this research project thus far. Fourth and finally, in the open discussion phase, participants who actively participated in the performance-experiment could share their experience with the spectators and talk directly with the research team (see Fig. 7 for visuals of each phase; video documentation of the event is available at https://bit.ly/Basic-Assumption).

Regarding the manipulation check. An independent-sample $t$-test revealed a significant effect of condition, $t(1, 170) = -2.21$, $p = 0.028$, $d = -0.34$, such that participants in the unlimited condition perceived empathy as less limited ($M = 5.01$, SD = 1.38) compared to participants in the control condition ($M = 4.49$, SD = 1.67).

To test whether the manipulation influenced empathic reactions, a repeated-measures ANOVA with the target identity (ingroup vs. outgroup) as a within-participant variable, and the condition (unlimited and control) as a between-participants variable revealed a significant Target identity × Condition interaction, $F(1, 170) = 5.09$, $p = 0.025$, $d = 0.35$ (Fig. 8). Whereas participants in the control condition felt more empathy toward the ingroup member than the outgroup member ($M = 4.89$, SD = 1.00; $M = 4.47$, SD = 1.12; $p < 0.001$), no

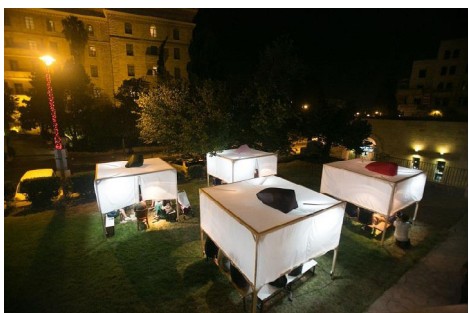
Phase 1: Introduction

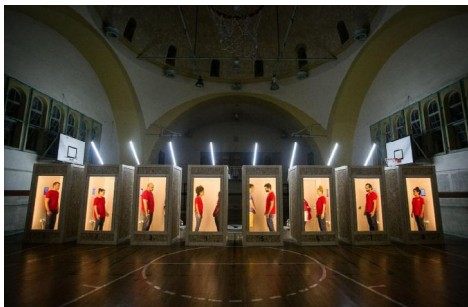
Phase 2: Intergroup encounter

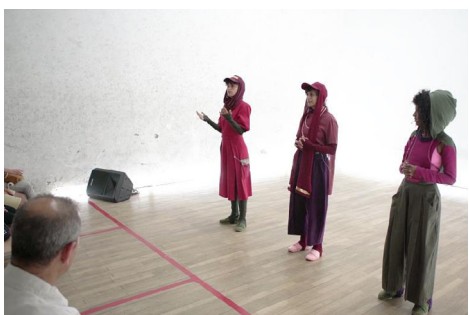
Phase 3: Science outreach and debriefing

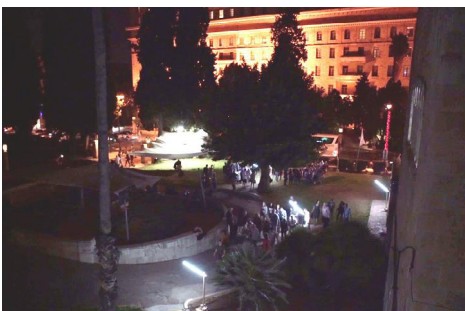
Phase 4: Open discussion

**Fig. 7 | The performance-experiment phases in Study 5.** The event included four main phases: (1) Introduction in which participants were randomly assigned to an experimental or a control condition; (2) Intergroup encounter in which participants met separately with an Arab actor and with a Jewish actor, each of whom shared a sad personal story with them; (3) Debriefing and science outreach in which all attendees watched a video art and a live stage performance sharing a review of the literature on outgroup empathy and the main findings of this research project thus far; and (4) Open discussion in which all attendees could share their experience with each others and talk directly with the research team.

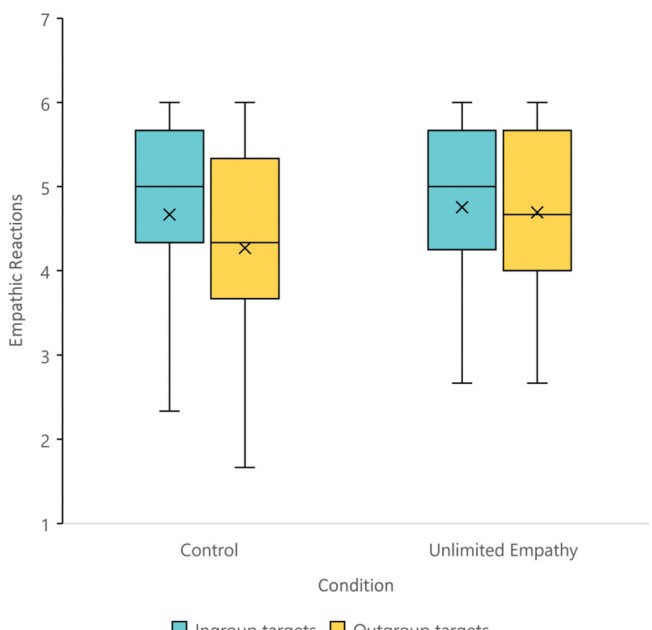

**Fig. 8 | Empathic reactions toward ingroup and outgroup targets, as a function of condition (i.e., unlimited empathy, _n_ = 88; and control, _n_ = 84).** A significant target identity × condition interaction, $F(1, 170) = 5.09$, $p = 0.025$, $d = 0.35$. A significant difference in empathy toward ingroup and outgroup members among participants in the control condition ($p < 0.001$). No significant difference in empathy towards ingroup and outgroup members among participants in the unlimited condition ($p = 0.344$). The plot presents the maximum and minimum values as whiskers, the interquartile range as the vertical length of each box, the mean as the x marker, and the median as the horizontal line within each box. All tests are two-tailed. Source data are provided as a Source Data file.

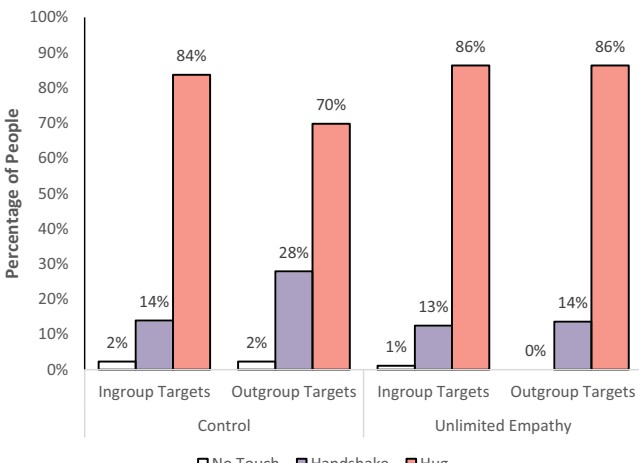

**Fig. 9 | Empathic behavior (no interpersonal touch; handshake; hug) toward ingroup and outgroup targets, as a function of condition (i.e., unlimited empathy, _n_ = 88; and control, _n_ = 86).** A significant difference in selecting the most empathic behavior (i.e., a hug) toward ingroup and outgroup members among participants in the control condition ($p < 0.001$). No significant difference in selecting the most empathic behavior toward ingroup and outgroup members among participants in the unlimited condition ($p = 1$; McNemar's chi-square test). All tests are two-tailed. Source data are provided as a Source Data file.

significant difference was found in empathy towards ingroup and outgroup members among participants in the unlimited condition ($M = 4.72$, SD = 1.09; $M = 4.62$, SD = 1.08; $p = 0.344$).

To test whether the manipulation also influenced empathic behavior, the participants' behavior was rated as a hug being the most empathic response, followed by a handshake, and finally, no interpersonal touch. Given that nearly none of the participants chose no touch (four participants; 2.3% of the sample), we compared the two remaining categories (a hug vs. handshake). A McNemar's chi-square test, which is used for paired nominal data[54], revealed that in the control condition, there were more participants who selected the most empathic behavior (i.e., a hug) toward the ingroup member than the outgroup member (84 vs. 70% respectively, $p < 0.001$). In contrast, in the unlimited condition, the same number of participants selected the most empathic behavior (i.e., a hug) toward both ingroup and outgroup members (both 86%, $p = 1$; Fig. 9). In addition, participants in the unlimited condition expressed significantly more interpersonal touch toward the outgroup member compared with those in the control condition (86 vs. 70% respectively, $p = 0.016$).

Study 5 replicated the results in Study 3 in face-to-face intergroup interactions. It showed that promoting the belief that empathy is unlimited led people not only to experience more empathy toward outgroup members but also to engage in more empathic behaviors. These results suggest that cultivating the belief that empathy is unlimited could decrease the intergroup empathy bias and its behavioral implications.

To examine the robustness of the results across all four experimental studies (Studies 2–5), we conducted a meta-analysis. The meta-analysis aggregated the effect sizes of the difference in outgroup empathy between people who were led to believe empathy is unlimited compared to those who led to believe empathy is limited or received no information. We meta-analyzed the studies using fixed

effects in which the mean effect size was weighted by sample size and comparing the unlimited condition with the limited or control conditions. Under these aggregated conditions, participants in the unlimited condition felt significantly greater empathic emotions (M $d = 0.34$, $Z = 4.04$, $p < 0.001$, 95% CI = [0.17, 0.50] compared with those in the limited or control conditions.

## Discussion
The main goal of the current research was to examine whether promoting the belief that empathy is unlimited can attenuate ingroup empathy bias, resulting in an increase in empathy and prosocial behaviors toward outgroup members. These hypotheses were tested using correlational and experimental designs, both in classic socio-psychological experiments and in two performance arts synthesized with psychological experiments. This concept enabled us to test our hypotheses in a real-world (albeit controlled) context with high external validity.

Our findings suggest that the belief about empathy as an unlimited resource is positively linked to empathy toward outgroup members, but rarely linked to empathy toward ingroup members. We found that the belief that empathy is an unlimited resource is associated with greater empathy and prosocial behavioral tendencies toward outgroup members and, consequently, with a decrease in intergroup empathy bias. This was true in various intergroup contexts, including different ethnic, national, religious, and political groups.

The suggested mechanism underlying these findings is that intergroup empathy bias is driven, in part, by people's tendency to believe that empathy is a limited resource. This belief leads to a zero-sum mindset[30] such that feeling empathy toward outgroup members seems to come at the expense of ingroup members. Given that people tend to favor their ingroup[55], they choose to preserve empathy for their own group and try to avoid feeling empathy toward outgroups. But when empathy is believed to be unlimited, zero-sum thinking is challenged. People are thus less concerned that by feeling empathy toward outgroup members, they would have less empathy for their ingroup.

The present research has both theoretical and practical implications for understanding and inducing outgroup empathy. First, previous research mainly investigated how situational factors influence

empathy toward others. For example, when people believe that one's misfortune is a result of external causes, rather than internal ones, they become more empathic to their situation[56]. People engage in more empathy-related behaviors and donate more generously to an identifiable victim than a non-identifiable victim[57]. People feel less empathy when they tend to believe it would lead to costly outcomes[58]. Furthermore, people feel more empathy and donate more to charities when they believe their group norms endorse such behavior[29,59]. These factors relate to the perception of the target in need and to the expected outcomes of feeling empathy. Our research contributes to the literature by pointing to a different factor, related to beliefs about empathy itself, that impacts empathic reactions.

Second, the belief about empathy as an unlimited resource uncovers a distinct mechanism from other lay theories about empathy. For example, the belief about the malleability of empathy[28,29] distinguishes between people with a fixed mindset of empathy who think that empathy is a stable trait, and people with a growth mindset who believe that empathy can be developed over time with effort. According to this theory, people with a growth mindset (as opposed to people with a fixed mindset) are typically more motivated by learning goals and may thus be more likely to regard the challenging intergroup context as an opportunity for growth. However, our research suggests a different mechanism in which the more people believe empathy is a limited resource, the more they think that empathy toward outgroup members comes at the expense of their own group members. Consequently, such individuals are more likely to empathize with ingroup members rather than with outgroup members. Whereas prior work (i.e., malleability beliefs) focused on the malleability of empathy, our work focuses on the distribution of empathy as a resource.

Third, empathy is often considered a reactive, uncontrolled response[60]. However, the current research suggests that empathy can be actively regulated in accordance with people's motivations[31,46]. People are motivated to regulate their emotions, in part, in order to attain group-related benefits[61], such as preserving resources for one's group. When people believe that empathy is limited, they may be more motivated to decrease empathy toward outgroup members because they are concerned it would come at the expense of their own group. In contrast, when people are led to believe that empathy is unlimited, the motivation to preserve empathy for the ingroup is rendered irrelevant, because there is sufficient empathy for everyone.

Lastly, prior research examined how intergroup relations are affected by the beliefs of zero-sum competition and zero-sum resources[34]. Such studies found that people who believed that the more an outgroup obtains, the less is available for their own ingroup, held more negative attitudes toward outgroup members[62]. While this research focused on zero-sum beliefs in general, the current research suggests that such beliefs about empathy itself are also essential for increasing outgroup empathy and prosociality.

From an applied perspective, the proposed intervention of changing beliefs about the limited nature of empathy has several benefits over existing interventions. Current interventions involve changing people's perceptions in a specific context. For example, perspective-taking interventions that ask people to take the point of view of the rival outgroup were found to increase empathy, intergroup helping, and recognition of ingroup misdeeds[63]. Other interventions increase empathy by changing perceptions of social norms regarding outgroup empathy[7,64]. While such interventions aim to increase empathy toward outgroups, the uniqueness of our intervention stems from the fact that it does not focus on any specific intergroup context. Rather, the intervention conveys a general message that can be applied in any situation. Trying to convince people directly that they can feel more empathy toward outgroup members is likely to provoke resistance, which this manipulation hopefully overcomes. This could be an advantage when applying the intervention in a variety of intergroup contexts without the need to adjust it to a certain context[65].

In addition, one of the main innovations of this project relies on the effectiveness of the intervention in realistic social contexts, such as those being tested in our research. Using art that generates powerful and meaningful experiences[66] could serve as a great platform to convey a counterintuitive psychological message (i.e., that empathy is unlimited) to the general population. Conveying such a message, in turn, could create a meaningful social change outside the laboratory. Participatory performance art that enables audiences to actively change and contribute to the art work[52] was a suitable platform for our psychological intervention for several reasons. Performance art brings people together, provides neutral spaces in which people can interact face-to-face, and even develop friendships. In addition, taking part in performance art projects can get people more involved in community issues, empowers and helps them gain control over their lives. Moreover, participating in performance art develops people's creativity and openness to new ideas[67], such as scientific discoveries.

Our intervention may have implications for other domains in which empathizing with outgroup members is needed. For example, this intervention could be used to train professionals who are expected to extend their empathy toward diverse groups and treat them equally. This can include professionals who work with heterogeneous groups and may be prone to discriminate against minorities[68], or professionals who are expected to treat all people equally regardless of their social background but often fail to do so[69]. This intervention may be useful in times of crisis (such as during global infectious diseases like the COVID-19 pandemic), because, in such times, the perceived scarcity of resources (e.g., jobs and health care services) leads to zero-sum biases and intergroup discrimination[32,70].

Together with these potential contributions, the present research also has several limitations that could potentially be addressed in future research. First, our research mainly focused on the emotional component of empathy (i.e., empathic concern) and related prosocial behaviors, but less on personal distress, and perspective-taking. Given that the subcomponents of empathy are deeply intertwined[71], we should expect the intervention to have a similar effect on all of them. However, some studies indicate that when empathy is expected to be emotionally exhausting[72] or cognitively effortful[73], people are less likely to feel empathy.

Second, it is important to continue investigating how changing the belief about empathy as an unlimited resource interacts with other motivational forces that lead to decreased empathy for outgroup members. For example, in zero-sum competitions (like sports games) in which one group's failure is tied to the other group's success, empathizing with outgroup members might be seen as antithetical to the pursuit of ingroup goals[31]. In such cases, even if people believe empathy is unlimited, they can still think that the outgroup suffering is a good thing for their own group. In addition, some prosocial behaviors might be more challenging, especially if they are objectively limited and perceived as zero-sum. For example, "costly" helping, such as donating money to outgroup members[74], might be less affected by the current intervention, because, for most people, money truly is a limited resource. In such cases, when there are two (or more) conflicting motives, the increased empathy might not be translated into prosocial action.

Third, the current study examined whether changing the belief about empathy as an unlimited resource alters people's level of empathy toward two to four individual people, either from the ingroup or from the outgroup, immediately after being exposed to the manipulation. It would be important to test the lasting effects of the current intervention and its effectiveness in the context of mass suffering in which people tend to feel less empathic emotions[75]. Fourth, the intervention was examined in an adult population. As social identity and group preferences develop at an early age[76], and prejudice peaks at the age of 5–7 years[77], the role of empathy and its development are crucial during these early ages and later in adolescence[78].

Thus, future research should examine whether the belief about empathy as unlimited can foster outgroup empathy among younger people. Lastly, the current project examined the effect of the intervention as part of performance art in the context of exhibitions and festivals. Although we managed to bring together people from diverse groups, these interactions differ from natural and spontaneous intergroup interactions in that the audience is already somewhat self-selected. As such, it would be important to examine the intervention in more natural settings.

In conclusion, the current project shows that the belief about empathy as an unlimited resource can facilitate intergroup empathy. Furthermore, it offers an intervention for reducing intergroup bias and increasing empathy toward outgroups, which are important in improving intergroup relations. We hope the current project will open the door for possible extensions that could increase empathy where it is needed most. Moreover, using the performative paradigm being developed in this project can be used as a way to conduct research in the social science fields, where the data collected is often insufficient for the understanding of complex phenomena[79]. Finally, we believe that multidisciplinary teams which collaborate and combine different disciplines would benefit both research and practice as large-scale societal problems cannot be solved by a single discipline[80].

## Methods

Our studies comply with all relevant ethical regulations. Studies 1–5 were approved by the Institutional Review Board (IRB) at Reichman University (former name: the Interdisciplinary Center), Herzliya. This IRB decided that the Pilot Study is exempt from their approval. We obtained informed consent from all participants for participation in the research and for publication of the images in Figs. 5 and 7, and in the video documentation from those presented there. In the Pilot Study, parental consent was obtained from the survey companies. As part of the statistical analyses, we tested that the required assumptions are met, including equality of variance and normality.

### Pilot study

As part of a nationwide education project, we collected a large sample of 1308 Israeli teenagers ($M_{age}$ = 16.65 years, SD = 2.662; 53.7% females) from three different social groups in Israel: Jewish Secular, Jewish Religious, and Arab. Participants were recruited via survey companies (i.e., iPanel and Geocartography) to complete an online survey about Israeli society in exchange for monetary compensation. The survey was administered in Hebrew and Arabic. Participants indicated their social identity ("Which of the following four groups do you most identify with? Secular Jews; Religious Jews; Ultra-Orthodox Jews; Arabs"), rated their empathic sentiments (from 1 = not at all, to 6 = very much) toward their ingroup and outgroup members ("To what extent do you feel empathy (understanding and sharing the feelings and thoughts of others) toward the following groups? Secular Jews; Religious Jews; Ultra-Orthodox Jews; Arabs"). In addition, they indicated their belief about empathy as an unlimited resource ("People have an infinite amount of empathy at their disposal") on a six-point scale (from 1 = not at all to 6 = very much).

### Study 1

We recruited 182 US participants ($M_{age}$ = 34.56 years, SD = 11.29, 57.7% females) via Amazon's Mechanical Turk who identified themselves as either liberals or conservatives (participants who identified as centrists with respect to their political ideology or held mixed political views were excluded). To obtain 80% power for a moderation model, we needed a sample size of 100 participants based on the lowest effect size found in the pilot study ($d$ = 0.2). Given the study aimed to examine an additional hypothesis (unrelated to the current project) that required more

participants, the final sample was almost doubled. In terms of political ideology, 52.7% of the participants self-identified as liberal and 47.3% of the participants self-identified as conservative. The study was conducted online at two different time-points. In the first assessment, participants indicated their political ideology (from 1 = very conservative, to 7 = very liberal) in four categories ("Many people use the terms "liberal" and "conservative" to identify different political views. Regarding your own political views, where do you place yourself in the following categories? "In general"; "Regarding economic issues (e.g., taxation, free markets, property rights, equality)"; "Regarding social issues (e.g., family values, religion, cultural traditions, minorities)"; "Regarding foreign policy and security" (e.g., military, international relations; α = 0.97). They also rated their belief about empathy as an unlimited resource on a scale from 1 = strongly disagree to 7 = strongly agree ("People have an infinite amount of empathy at their disposal"; "There is no limit to the extent of empathy a person can feel"; "There is a limit to how much empathy we can feel towards others"; "One cannot be empathetic with everyone"; last two items are reversal; α = 0.75). Finally, participants provided demographic information.

In the second assessment, which was administered a week later, participants were randomly assigned to one of two target conditions (i.e., political ingroup, political outgroup). Participants read a bogus empathy-inducing article about 12 people, described as either liberals or conservatives, who were badly injured in a protest that was overcrowded (for the full articles, see Supplementary Fig. 1). After reading the article, participants rated their empathic reactions (from 1 = not at all, to 7 = very much) toward the injured people ("Following the article, to what extent do you currently feel each of the feelings below? Empathy; Sympathy; Compassion"; α = 0.94). Note that data was collected as part of a larger study that included additional measures and another condition that is less relevant to the current study (i.e., neutral group). Part of the data was published in ref. 46.

### Study 2

To obtain 80% power for a mixed design with both within- and between-participant factors, we needed a sample size of 194 participants based on the lowest effect size found in the pilot study ($d$ = 0.2). We slightly over-sampled to account for possible attrition. The study was pre-registered on April 26, 2018 (https://aspredicted.org/6h6a8.pdf). Two hundred and six US participants ($M_{age}$ = 33.5 years, SD = 11.5, 57.5% females) were recruited via Amazon's Mechanical Turk (six participants who failed the attention check were excluded from the analysis). They were randomly assigned to one of two conditions—limited empathy ($n$ = 99), or unlimited empathy ($n$ = 101). In each condition, participants were presented with the definition of empathy-based on [81] (Supplementary Fig. 2). Then, participants in the unlimited (vs. limited; appears in brackets) condition read that "Recent studies have found that empathy is an unlimited [a limited] resource so people can [cannot] feel it toward a large number of people". To reinforce the manipulation, participants in the unlimited (vs. limited) condition were asked the following: "Imagine that you are about to meet people in distress. Toward how many of them could you feel empathy on a scale from 0—can't feel empathy toward anyone; To 300 [3]—can feel empathy toward three hundred [three] people?" (Supplementary Fig. 2). Because people tend to rate their positive traits above average[82], most participants (74.8%) in the unlimited condition indicated they can empathize with 150 people (mid-scale) or more. Following the manipulation, participants read four empathy-inducing testimonies based on real stories of Syrian refugees who were tortured in their homeland. Testimonies were presented in a counterbalanced order. After reading each testimony, participants rated their empathic reaction on a scale of 1 = not at all, to 7 = very much ("Following the story, to what extent do you currently feel empathy toward

[Syrian refugee name]?"). Finally, participants completed a manipulation check, as in Study 1.

## Study 3

One hundred sixty-two participants who self-identified as Israeli Jews ($M_{age}$ = 37.33 years, SD = 12.97, 52.7% females) were recruited via a survey company (i.e., iPanel). The sample size was determined based on the effect size found among the Jewish sample in the pilot study ($d$ = 0.37). To obtain 80% power, we needed a sample size of 144 participants. We slightly over-sampled to account for possible attrition (twelve participants who failed the attention check were excluded from the analysis). The study was pre-registered on August 5, 2017 (https://aspredicted.org/fj663.pdf). Participants were randomly assigned to one of three conditions–limited empathy ($n$ = 51), unlimited empathy ($n$ = 52), and control ($n$ = 47). In each condition, participants were presented with information about the definition of empathy (same as in Study 2) and whether it is an unlimited or limited resource, or with no information regarding its limitation (i.e., control; Supplementary Fig. 3). Following the manipulation, participants read two empathy-inducing articles about a family in need (health or financial issues; Supplementary Fig. 4). One of the articles described a Jewish family (i.e., ingroup) and the other article described an Arab family (i.e., outgroup). The type of issue (financial or health) and identity of the family were counterbalanced. After reading each story, participants rated their empathic reaction on the scale used in Study 1 (α (regarding the first article) = 0.85; α (regarding the second article) = 0.91). They also rated their support of prosocial actions on a scale from 1 = not at all, to 7 = very much ("Support the authorities provide help to this family"; "Promoting policies that improve the condition of families in a similar situation"; "Assistance of volunteer organizations"; "Contacting the relevant Minister to handle such cases"; α (regarding the first article) = 0.87; α (regarding the second article) = 0.91). Finally, participants completed a manipulation check, using the scale of belief about empathy as an unlimited resource used in Study 1 (α = 0.83).

## Study 4

A priori power analyses based on the effect size of empathic reactions in Study 2 lead to unacceptably small sample sizes (i.e., less than 15 participants). Therefore, we aimed to recruit at least 100 participants (50 in each of the two conditions). The study was pre-registered on May 9, 2018 (https://aspredicted.org/7cy6n.pdf). US participants ($M_{age}$ = 34.8 years, SD = 11.5, 55.6% females) attended the performance art-experiment that took place in a performance venue in Chicago. On arrival, the 108 participants were randomly assigned to one of two conditions–limited empathy ($n$ = 53) and unlimited empathy ($n$ = 55). The manipulation and measures were the same as in Study 2 except that they were performed and asked by actresses and not in an online survey.

## Study 5

A priori power analyses based on the effect size of empathic reactions in Study 3 led to a very small sample size (i.e., less than 50 participants). Therefore, we aimed to recruit at least 100 participants (50 in each of the two conditions). However, given that the study was part of an open perforce art, we accepted more participants to take part in the event. The event was advertised under the name "Basic Assumption—a performance-experiment" without mentioning that it was about empathy or intergroup relations in order to avoid sampling bias. Among the people who registered and arrived at the event (-700), we randomly selected 176 participants who self-identified as Israeli Jews to take an active part in the performance-experiment ($M_{age}$ = 26.84 years, SD = 13.89, 50.3% females). The others watched the performance-experiment as part of the audience. Two participants could not rate their empathic feelings due to technical issues, and two other participants chose not to participate. Participants registered to different timeslots throughout the day, such that every 20 min, a group of eight participants began the performance art-experiment. Participants were randomly assigned to an experimental ($n$ = 88) or a control condition ($n$ = 86). Due to limitations regarding the number of participants, we did not include a condition that manipulates the belief about empathy as limited (which showed similar effects in Study 3 as the control condition). First, all participants met an actress who welcomed them and informed them that in the performance they would learn about empathy. Actresses explained what empathy is (an elaboration of the definition presented in Study 3). In addition, in the experimental condition, participants were provided with additional information that altered their belief about empathy as an unlimited resource. Participants were introduced to results of a study that found that people who express more empathic behaviors in the workplace (such as taking the time to listen to coworkers) also feel more empathy toward their family members (This information was based on a study that actually found the opposite association; 31). Then, each participant was asked to enter a separate cabin without any additional information about the next phase of the performance-experiment. A few seconds later, actors entered each of the cabins. To avoid gender-related biases, we assigned participants to actors of the same gender. Each actor shared with the participants a personal 3-min empathy-inducing story (an elaboration of the stories presented in Study 3). In the end, empathic behavior was measured by the participants' response to the actors' question, whether they wanted to end the encounter as they are (with no interpersonal touch), with a handshake, or with a hug. Then, the actors left their cabins and closed the doors. Participants were then instructed to rate their level of empathic reaction (Same as in Study 3; α (regarding the first story) = 0.73; α (regarding the second story) = 0.78). After a few minutes, another actor entered the cabin, shared her story, and repeated the same procedure. Half of the participants first met an Arab actor and then a Jewish actor, and the other half met the actors in the opposite order. Finally, participants indicated their belief about empathy as an unlimited resource, as a manipulation check (Same as in Study 1; α = 0.79).

## Reporting summary

Further information on research design is available in the Nature Portfolio Reporting Summary linked to this article.

## Data availability

All data analyzed during the current studies are available at Open Science Framework, https://doi.org/10.17605/OSF.IO/4U26S[83]. Source data are provided with this paper.

## Code availability

The code for analysis is available at https://doi.org/10.17605/OSF.IO/4U26S[83].

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

## Acknowledgements

We thank the following organizations and individuals for their help in carrying out the studies. Pilot Study—Data were collected as part of a national project by "aChord: Social Psychology for Social Change" at the Hebrew University. Study 4—As Much As You Want / Chicago, USA—Curator: Megha Ralapati; Performers: Risha Tenae, Charm Tims, Margaret Morris, Eliza Myrie; Documentation: Chuck Przybyl; Set Fabrication: Peter Reese; Organized by: Hyde Park Art Center; Venue: Links Hall. Study 5 —Basic Assumption/Jerusalem, Israel—Directors of Mekudeshet: Naomi Bloch Fortis, Itay Mautner; Producer: Omer Alsheich; Costumes: Hila Shapira; Supporting performers: Students of School of Visual Theater, Bezalel Academy of Arts and Design and Sapir College; Location: YMCA, Jerusalem; [Performance credits] Performers: Bar Altaras, Sahar Damoni, Tovit Semay, Adi Nizan, Anat Federschneider, Bahat Calatchi, Hana Vazana Grunwald, Bar Elyakim, Rotem Goldenberg, Mouna Hawa, Lamis Amar, Omer Perlman Striks, Gassan Ashkar, Hanan Asraf, Murad Hassan; Space Design: Shay Id Aloni; Light Design: Yair Vardi; Photographer: Michal Fattal; [Video credits] Performers: Bar Altaras, Sahar Damoni, Tovit Semay; Producer, Shooting Days: Sharon Daniel; Cinematography: Avigail Sperber; Second Photographer: Amit Chachamov; Editing and Post: Hinda Weiss; Sound Editing: Lukas Turza; Lighting Technician: Ging; Audio Technician: Amos Zipori; Hair and Makeup: Ofer Ben Natan; Assistant Photographer: Arkady Ostrovsky; Lighting Assistant: Nir Ish-Shalom; Carillon player: Prof. Gaby Shefler; Subtitles: Inbar Hagai; Teleprompter: Yael Shani; Translator: Kim Weiss and the Mekudeshet translation team; Arabic Editor: Riman Barakat; Equipment Rental: Utopia Camera Services Ltd. This work was supported by the following fundings: A grant from the European Research Council [864347] (E.H.), a collaborative research grant from IDC Herzliya and Bezalel Academy of Art Jerusalem (E.H. and E.A.), grants from Artis (E.A.), The Arison Family Foundation (E.A.), CEC ArtsLink Independent Projects award (E.A.), Israeli Pais Art and Culture Council (E.A.), and a grant from the Levy Eshkol Institute for Economic, Social and Political Research, the Hebrew University (Y.H.).

## Author contributions

Y.H., E.A., and E.H designed and performed the research and art-performance; Y.H. first accessed the data and analyzed it; Y.H. wrote the main parts of the paper; E.A., E.H., M.T., and D.S.-S. were involved in the writing process and provided critical revisions, and E.A. directed all performance art aspects.

## Competing interests

The authors declare no competing interests.
