## [Peer Review File · Nature Communications]

Using performance art to promote intergroup prosociality by cultivating the belief that empathy is unlimitedREVIEWER COMMENTS

Reviewer #1 (Remarks to the Author):

In this manuscript, the authors provide evidence that greater belief in “empathy as a limited resource” is associated with lower intergroup empathy. They demonstrate the efficacy of a novel (and commendable) performance art intervention to increase intergroup empathy (via enhancing belief in empathy as an unlimited resource). Further, they provide conceptual replications and demonstrate the robustness of their effects across 6 studies (total N=2,118) by: 1) recruiting diverse samples that enable them to distinguish intergroup boundaries via different aspects of group identity; and 2) employing diverse measures of intergroup empathy. This is timely and important work, and the strength and novelty of their performance art intervention alone is sufficient to warrant its publication. My only major concern is the authors’ clarity in their treatment of the central concept of empathy; both with regard to their usage and how they operationalize it, as well as their contextualizing the present work in the literature. I also have some minor concerns about the authors analyses and their reporting of results. Finally, I will assert that the authors in fact *undersell* the novelty and contribution of their performance art intervention, and I encourage them to highlight this at greater length. I believe addressing these concerns will help the strengths of the manuscript stand out in starker relief and catalyze its contribution to the literature.

- Major issues

- o Clarifying the concept of “empathy”

☐ Considering that it is the authors’ primary construct of interest, it is surprising to see a relative lack of clarity in discussing it. Consider the following instances (and my thoughts on reconciling these below):

1. L.46-8: “Outgroup empathy – understanding and sharing the emotions of people who do not belong to one’s social group, and having feelings of sympathy and compassion for those in need...”
2. Figure S2: “Empathy is defined as the ability to understand and share the feelings and thoughts of others. For example, empathizing with someone in distress involves understanding the situation from his/her perspective and feeling his/her negative emotions.”
3. L.573-5: “First, our research mainly focused on the emotional component of empathy (i.e., empathic concern) and related prosocial behaviors, but less on personal distress, and perspective taking.”
4. L.655-6 (Methods for Studies 1,3, and 5): “...to what extent do you currently feel each of the feelings below? Empathy; Sympathy; Compassion)”
5. L.677-9 (Methods for Studies 2 and 4): “...to what extent do you currently feel empathy toward []?”

☐ (1) contains all of the threads of the knot the authors need to untangle.

- The authors begin with the compound concept “outgroup empathy.” triangulating across (1) and (2), I think the authors might define “empathy” as “understanding and sharing the feelings and thoughts of others.” I think the authors mean to apply this concept to an “outgroup”; from (1), I think they would say this means people “who do not belong to one’s social group.” This is the operationalization the authors seem to employ in their dependent measure in Studies 2 and 4; (5) above.

- Yet, the concluding clause of (1)—“and having feelings of sympathy and compassion for those in need”—introduces two affective states (i.e. sympathy and compassion) that the authors operationalize in their dependent measure in Studies 1, 3, and 5; (4) above.

- Further, although I agree with the claim they make in (3), this seems inconsistent with their assertion in (2). (Additionally, they appear to be referencing work by Davis, citation below, which I believe they should cite.)

- Finally, the inclusion of “...for those in need” (1) implies targets that are in perceptible distress; if the authors feel this qualifier is necessary, more elaboration is needed.

☒ Related to these concerns, their dependent measure in Study 5 (i.e. L.433-4, “no interpersonal touch, with a handshake, or with a hug”) is problematic.

- First, they employ (L.461-2) “a McNemar’s chi-square test which is used for paired nominal data.” Unless I am mistaken, I believe the authors are investigating a 3x3 contingency table.

- Also, while this variable is clearly ordinal, it is not justified to treat it as interval. If the authors wish to treat it as such, I would suggest a short (online, if possible for convenience) study to obtain ratings of each of the three values of this variable on a continuous measure of empathy.

☒ In sum, I strongly encourage the authors to clarify their definition and usage of the term “empathy.” I include the following citations as a guide in case they are instructive:

- Cuff, B. M., Brown, S. J., Taylor, L., & Howat, D. J. (2016). Empathy: A review of the concept. *Emotion review*, 8(2), 144-153.

- Davis MH. Measuring individual differences in empathy: Evidence for a multidimensional approach. *Journal of Personality and Social Psychology*. 1983; 44(1):113.

- Jordan, M. R., Amir, D., & Bloom, P. (2016). Are empathy and concern psychologically distinct? *Emotion*, 16(8), 1107.

- Shamay-Tsoory SG, Aharon-Peretz J, Perry D. Two systems for empathy: a double dissociation between emotional and cognitive empathy in inferior frontal gyrus versus ventromedial prefrontal lesions 2009-03-01 00:00:00. 617±27 p.

- Wispe L. The distinction between sympathy and empathy: To call forth a concept, a word is needed. *Journal of personality and social psychology*. 1986; 50(2):314.

- Zaki, J. (2014). Empathy: a motivated account. *Psychological bulletin*, 140(6), 1608. (cite 30)

o Contextualizing the present work in the literature

☒ The authors should situate their work within other literature documenting interventions of intergroup empathy, for example:

- Čehajić-Clancy, S., Goldenberg, A., Gross, J. J., & Halperin, E. (2016). Social-psychological interventions for intergroup reconciliation: An emotion regulation perspective. *Psychological Inquiry*, 27(2), 73-88.
- Paluck, E. L. (2009). Reducing intergroup prejudice and conflict using the media: a field experiment in Rwanda. *Journal of personality and social psychology*, 96(3), 574.
- Stephan, W. G., & Finlay, K. (1999). The role of empathy in improving intergroup relations. *Journal of Social issues*, 55(4), 729-743.
- Vanman, E. J. (2016). The role of empathy in intergroup relations. *Current Opinion in Psychology*, 11, 59-63.

☒ Although I agree with the following claim, it should be supported with citations: L.49 “Despite its obvious benefits in promoting prosocial behavior between groups...”

☒ The authors make the claim that empathy is a limited resource (L.60-2) but do not support this claim until L.79-86 (citations 22-6); I believe these lines should follow L.62. Additionally, the authors may wish to consult the following citations to further bolster their claim:

- Bloom, P. (2017). *Against empathy: The case for rational compassion*. Random House.
- Cameron, C. D., & Payne, B. K. (2011). Escaping affect: how motivated emotion regulation creates insensitivity to mass suffering. *Journal of personality and social psychology*, 100(1), 1.
- Västfjäll, D., Slovic, P., Mayorga, M., & Peters, E. (2014). Compassion fade: Affect and charity are greatest for a single child in need. *PloS one*, 9(6), e100115.

- Minor issues

o The authors emphasize performance art in the Discussion (L.549-61), though I would encourage them to discuss it at greater length (earlier, and throughout) because it is the most novel component of their investigation and it is noteworthy and very interesting.

o This is entirely at the discretion of the authors, but I would encourage them to consider conducting a meta-analysis of the studies they present within this manuscript. I believe such an analysis would bolster their presentation of the robustness of their results, as well as providing a compelling figure to readers.

o It would be helpful to readers to include the number of studies and the total sample size (by my count: 2,118; 1308+182+194+150+108+176) in the abstract (and perhaps also towards the end of the introduction, e.g. L.129).

o This paper has numerous grammatical, syntax, and miscellaneous language issues which need to be addressed. For example, L.41-2 is an incomplete sentence and the verb conjugation (“force”) is incorrect:

☒ “From the beginning of the 21st century, armed intergroup conflicts cost the lives of more than 1 million people worldwide (3), force more than 70 million people to flee their homes (4).”

o General comments on analyses and results

☒ All figures should represent variance with error bars (currently missing from Figure 1). It is somewhat a question of style, though I think it is becoming more normative and I think it is more helpful to include 95% CI rather than SE.

☒ I appreciate that the authors have included methodological details at the end of the manuscript (as per journal customs), though it would help readers to include basic details about their dependent measures throughout the manuscript (e.g. for Study 1, consider inserting the following text from Methods around L.180):

- L.654-57 “After reading the article, participants rated their empathic reactions (from 1 = not at all, to 7 = very much) toward the injured people (“Following the article, to what extent do you currently feel each of the feelings below? Empathy; Sympathy; Compassion”; $\alpha = .94$).)”

☒ Although the authors detail their power calculations in the Methods, they should acknowledge that they computed these for each study early in the manuscript (and direct readers to the Methods for more detail).

☒ In the Methods (e.g. Study 5), the authors provide separate Cronbach’s alphas for their multi-item dependent measures for each article. Perhaps I am ignorant of some methodological developments, but it is my impression that alphas should be calculated *across* condition, not within them. Therefore, one alpha value is sufficient (and should be presented in the main text, rather than the Methods, as indicated above).

☒ This is a very minor style note on presenting condition means, but it would clarify the authors’ writing to comprehensively check their manuscript to make the following edit:

- For example, L.446-448: “Such that participants in the unlimited condition perceived empathy as less limited compared to participants in the control condition (M = 5.01, SD = 1.38; M = 4.49, SD = 1.67 respectively).”
- “Such that participants in the unlimited condition perceived empathy as less limited (M = 5.01, SD = 1.38) compared to participants in the control condition (M = 4.49, SD = 1.67).”

o Pilot study

☒ Table 1 documents (L.157) “A series of simple bivariate correlations (with no correction)” but it seems correction is warranted here because it presents multiple comparisons (i.e. empathic sentiments toward different groups with belief in empathy as a limited resource).

☒ Figure 1 is somewhat misleading; they artificially discretize “belief about empathy” (x-axis) yet plot lines which imply that points along the line are meaningful (which they are not in this context). Given that the authors are presenting the relationship between two continuous variables, why not superimpose regression lines (as they have done) over a scatterplot of the data?

o Study 1

☒ Figure 2 (as well as Figure 6 in Study 4)

- The data labels are confusing, in that it is unclear whether they index content or order (although they seem to imply the latter). In Figure S3, the authors present 4 testimonies (unlabeled) and write (L.231-2) that “all testimonies were presented in a counterbalanced order.” Therefore, does the label, e.g., “1st testimony” refer to “Alaa, 32, Daraya” or to the first testimony participants read?

- Also, the analysis the authors present is interesting, it is unclear how it bears on their hypotheses; why do you think participants became *more* empathic over the course of the “limited empathy” treatment? I would be inclined to keep this analysis—and to speculate on its implications—but I would not make this figure based on it.

- Instead, I would encourage the authors to test the interaction of testimony *content* and condition on empathic reactions. Presumably, the authors do not think that there is an effect of testimony content, and finding that this interaction is not significant (which would be my guess) would enable them to collapse the data across this dimension (making both figures more interpretable).

o Study 3

☒ The authors present two separate analyses for their “empathic reactions” and “support for prosocial actions” dependent measures; yet, because these represent multiple observations from the same participants, a more appropriate analysis would employ multivariate regression.

☒ I appreciate the authors’ use of the Hayes PROCESS models, though I encourage them to ensure that they are implementing their mediation model correctly here, given that they employ a categorical independent variable (see citation below for guidance).

- Hayes, A. F., & Preacher, K. J. (2014). Statistical mediation analysis with a multicategorical independent variable. *British journal of mathematical and statistical psychology*, 67(3), 451-470.

o Study 4

☒ I believe there is a typo in the authors’ reporting of means here:

- L.389-91 “On average, participants in the unlimited condition felt more empathy toward the refugees ($M = 5.5$, $SD = 1.25$), compared to participants in the limited condition ($M = 6.08$, $SD = 1.13$).”

Reviewer #2 (Remarks to the Author):

I've now read "Believing empathy is unlimited: Using performance art to reconstruct intergroup empathy and bring people together" with great interest. The work has several important strengths. First, it leverages a novel theoretical approach to intervening on empathy. Second, it offers a rigorous test of its hypotheses across several contexts, including multiple lab and field experiments. Third, it offers a cost-effective and scalable tool to address conflict. Finally, the manuscript is well written and easy to follow. In short, there was a lot to like here, and my overall evaluation was very positive.

I had the following questions and concerns:

1. The authors assert that changing perceptions of empathy-related capacities could facilitate intergroup empathy. This is because some individuals consider empathy a limited resource, which they choose to save for ingroup members. I agree with the authors that this is a critically important area for inquiry. However, I wonder how the authors think about how beliefs about limitations of empathy—the focus of the present intervention—interact with other motivational forces that deter empathy for outgroup members.

For example, empathizing with outgroup members is seen by some as antithetical to the pursuit of group goals, especially in instances of (real or perceived) zero-sum competition. Empathizing with outgroup members may also be more difficult than empathizing with ingroup members due to perceived dissimilarities, and it could therefore be seen as more costly than empathizing with ingroup members [1,2]. It would be informative to include some discussion on how beliefs about capacity interact with other empathic "avoidance motives", such as potential interference with group goals, or the motivation to avoid costly empathizing. This could fit nicely with a related point the authors make about limitations of the intervention in the discussion (lines 583- 588).

2. Along these lines, the manner in which the authors discuss "zero-sumness" related to empathy differs somewhat from how it is discussed by other researchers. Prior theoretical accounts explain how zero-sumness of resources may drive intergroup empathy differences, because individuals are motivated to amass resources for their ingroup. However, here the authors describe empathy itself as zero-sum, irrespective of resource limits (line 71). How do the authors imagine this paradigm to work in circumstance where resources remain zero-sum? Isn't it possible that people believe empathy is unlimited but still choose not to empathize with outgroup members? Why should changing this belief that empathy is unlimited help individuals overcome other empathic avoidance motives? This feels like an important theoretical point, and I would appreciate seeing this described in the introduction.

3. Although group-based differences emerge clearly in every experiment, the absolute levels of empathy for outgroup members were higher than what I would have expected based on previous work (especially

studies 2 and 4). I would appreciate it if the authors could offer comment on this, particularly because other studies find low levels of empathy for outgroup members (or even counter-empathy, such as schadenfreude) [3].

4. I wonder whether the authors have any thoughts on the different aspects of the performance art intervention offering both first-person and third-person insights into others' experiences. Previous work finds that "imagine-self" and "imagine-other" perspective taking instructions have very different consequences [4]. From my read, the performance art intervention included experiences consistent with each of these types of perspective taking. It would be helpful to see a brief discussion addressing this aspect of their intervention.

5. The authors describe collecting measures of sympathy and compassion for ingroup and outgroup members in addition to differences in empathy. Did the authors provide participants definitions of these terms, or otherwise distinguish empathy, sympathy and compassion? Were there group-based differences in sympathy and compassion ratings as well?

6. Were there any data exclusions in any of these experiments? If not, please state that explicitly.

In sum, this was a huge undertaking on the part of the researchers. I believe this article is of great theoretical and practical importance. It is an excellent piece of scholarship and with some revisions I hope to see it published at Nature Communications.

References

1. Zaki, J. (2014) Empathy: A motivated account. *Psychol. Bull.* 140, 1608–1647.
<https://doi.org/10.1037/a0037679>

2. Zaki, J. and Cikara, M. (2015) Addressing Empathic Failures. *Curr. Dir. Psychol. Sci.* 24, 471–476.
<https://doi.org/10.1177/0963721415599978>

4. Cikara, M. et al. (2014) Their pain gives us pleasure: How intergroup dynamics shape empathic failures and counter-empathic responses. *J. Exp. Soc. Psychol.* 55, 110–125. <https://doi.org/10.1016/j.jesp.2014.06.007>

5. Lamm, C. et al. (2007) The neural substrate of human empathy: effects of perspective-taking and cognitive appraisal. *J. Cogn. Neurosci.* 19, 42–58. <https://doi.org/10.1162/jocn.2007.19.1.42>

Reviewer #3 (Remarks to the Author):

A series of correlational and experimental studies are reported showing that leading participants to believe that empathy is unlimited (compared to limited) increased outgroup empathy.

This is a great paper and I enjoyed reading it. The theorising is sound, the studies are cleverly designed and competently conducted, producing noteworthy results. I particularly like the range of different groups and different conflicts that were studied including different ethnic groups (Jews vs. Arabs, Syrian refugees). Most impressive was the final study including the manipulation through performance and the study of behaviours. I would love to see more studies like this.

I can't think of ways to improve this paper and think it is an important paper that holds great promise to move the field of reducing intergroup conflict forward in important ways. The limited versus unlimited empathy concept is novel and has great potential practically.

I soon hope to see this paper in print in Nature Communications.

In what follows, we address all the comments in the order in which they appeared:

Reviewer #1

- 1. In this manuscript, the authors provide evidence that greater belief in “empathy as a limited resource” is associated with lower intergroup empathy. They demonstrate the efficacy of a novel (and commendable) performance art intervention to increase intergroup empathy (via enhancing belief in empathy as an unlimited resource). Further, they provide conceptual replications and demonstrate the robustness of their effects across 6 studies (total N=2,118) by: 1) recruiting diverse samples that enable them to distinguish intergroup boundaries via different aspects of group identity; and 2) employing diverse measures of intergroup empathy. This is timely and important work, and the strength and novelty of their performance art intervention alone is sufficient to warrant its publication. My only major concern is the authors’ clarity in their treatment of the central concept of empathy; both with regard to their usage and how they operationalize it, as well as their contextualizing the present work in the literature. I also have some minor concerns about the authors analyses and their reporting of results. Finally, I will assert that the authors in fact *undersell* the novelty and contribution of their performance art intervention, and I encourage them to highlight this at greater length. I believe addressing these concerns will help the strengths of the manuscript stand out in starker relief and catalyze its contribution to the literature.**

Thank you for the positive feedback!

Clarifying the concept of “empathy”

- 2. Considering that it is the authors’ primary construct of interest, it is surprising to see a relative lack of clarity in discussing it. Consider the following instances (and my thoughts on reconciling these below):**
 - (1) L.46-8: “Outgroup empathy – understanding and sharing the emotions of people who do not belong to one’s social group, and having feelings of sympathy and compassion for those in need...”**
 - (2) Figure S2: “Empathy is defined as the ability to understand and share the feelings and thoughts of others. For example, empathizing with someone in distress involves understanding the situation from his/her perspective and feeling his/her negative emotions.”**
 - (3) L.573-5: “First, our research mainly focused on the emotional component of empathy (i.e., empathic concern) and related prosocial behaviors, but less on personal distress, and perspective taking.”**
 - (4) L.655-6 (Methods for Studies 1,3, and 5): “...to what extent do you currently feel each of the feelings below? Empathy; Sympathy; Compassion)”**

(5) L.677-9 (Methods for Studies 2 and 4): “...to what extent do you currently feel empathy toward []?”

(1) contains all of the threads of the knot the authors need to untangle.

- **The authors begin with the compound concept “outgroup empathy.” triangulating across (1) and (2), I think the authors might define “empathy” as “understanding and sharing the feelings and thoughts of others.” I think the authors mean to apply this concept to an “outgroup”; from (1), I think they would say this means people “who do not belong to one’s social group.” This is the operationalization the authors seem to employ in their dependent measure in Studies 2 and 4; (5) above.**
- **Yet, the concluding clause of (1)—“and having feelings of sympathy and compassion for those in need”—introduces two affective states (i.e. sympathy and compassion) that the authors operationalize in their dependent measure in Studies 1, 3, and 5; (4) above.**
- **Further, although I agree with the claim they make in (3), this seems inconsistent with their assertion in (2). (Additionally, they appear to be referencing work by Davis, citation below, which I believe they should cite.)**
- **Finally, the inclusion of “...for those in need” (1) implies targets that are in perceptible distress; if the authors feel this qualifier is necessary, more elaboration is needed.**

We thank the reviewer for helping us to clarify the term ‘empathy’ as mentioned throughout the manuscript. Following the reviewer’s recommendations, we first define empathy as “understanding and sharing the feelings and thoughts of others”. Second, we define outgroup empathy as “empathy towards people who do not belong to one’s social group”. Third, as the reviewer pointed out, we adopted Davis’ definition which focuses on empathic concern as part of the emotional component of empathy, which involves feeling sympathy and compassion (Davis, 1983). As the reviewer recommended, we now cite Davis when defining the term empathy. Lastly, we changed the phrase “for those in need” to “for others’ suffering” as part of the definition of empathic concern. Changes in lines 49-57.

- 3. Related to these concerns, their dependent measure in Study 5 (i.e. L.433-4, “no interpersonal touch, with a handshake, or with a hug”) is problematic.**
 - **First, they employ (L.461-2) “a McNemar’s chi-square test which is used for paired nominal data.” Unless I am mistaken, I believe the authors are investigating a 3x3 contingency table.**
 - **Also, while this variable is clearly ordinal, it is not justified to treat it as interval. If the authors wish to treat it as such, I would suggest a short (online, if possible for convenience) study to obtain ratings of each of the three values of this variable on a continuous measure of empathy.**

The reviewer commented that McNemar's chi-square test is used for paired nominal data and not for variables with more than two categories like our dependent measure. Indeed, the original dependent variable consists of three categories (i.e., a hug, handshake, and no

interpersonal touch). However, given that nearly none of the participants chose no touch (four participants; 2.3% of the sample), we compared the two remaining categories (a hug vs. handshake). In such case, the McNemar's chi-square test is an appropriate analysis for comparing two categories. We hope this explanation satisfies the reviewer and we now describe this procedure more clearly in the manuscript.

- 4. In sum, I strongly encourage the authors to clarify their definition and usage of the term “empathy.” I include the following citations as a guide in case they are instructive:**
- **Cuff, B. M., Brown, S. J., Taylor, L., & Howat, D. J. (2016). Empathy: A review of the concept. *Emotion review*, 8(2), 144-153.**
 - **Davis MH. Measuring individual differences in empathy: Evidence for a multidimensional approach. *Journal of Personality and Social Psychology*. 1983; 44(1):113.**
 - **Jordan, M. R., Amir, D., & Bloom, P. (2016). Are empathy and concern psychologically distinct? *Emotion*, 16(8), 1107.**
 - **Shamay-Tsoory SG, Aharon-Peretz J, Perry D. Two systems for empathy: a double dissociation between emotional and cognitive empathy in inferior frontal gyrus versus ventromedial prefrontal lesions. *Journal of Personality and Social Psychology*. 2009; 96(3):563-574.**
 - **Wispe L. The distinction between sympathy and empathy: To call forth a concept, a word is needed. *Journal of personality and social psychology*. 1986; 50(2):314.**
 - **Zaki, J. (2014). Empathy: a motivated account. *Psychological bulletin*, 140(6), 1608. (cite 30)**

We thank the reviewer for providing these important references. We clarified the definition of empathy, by relying on most of the suggested citations including Cuff et al, 2016; Davis, 1983; Shamay-Tsoory et al., 2009; Zaki, 2014 (lines 49-57).

Contextualizing the present work in the literature

- 5. The authors should situate their work within other literature documenting interventions of intergroup empathy, for example:**
- **Čehajić-Clancy, S., Goldenberg, A., Gross, J. J., & Halperin, E. (2016). Social-psychological interventions for intergroup reconciliation: An emotion regulation perspective. *Psychological Inquiry*, 27(2), 73-88.**
 - **Paluck, E. L. (2009). Reducing intergroup prejudice and conflict using the media: a field experiment in Rwanda. *Journal of personality and social psychology*, 96(3), 574.**
 - **Stephan, W. G., & Finlay, K. (1999). The role of empathy in improving intergroup relations. *Journal of Social issues*, 55(4), 729-743.**
 - **Vanman, E. J. (2016). The role of empathy in intergroup relations. *Current Opinion in Psychology*, 11, 59-63.**

Following the reviewer's suggestion, we further elaborated on existing interventions of

intergroup empathy, citing the work the reviewer suggested, while also highlighting the added value of our intervention (lines 637-651).

6. **Although I agree with the following claim, it should be supported with citations: L.49 “Despite its obvious benefits in promoting prosocial behavior between groups...”**

Following the reviewer’s comment, we supported our claim by citing Halperin (2016) “Emotions in Conflict: Inhibitors and Facilitators of Peace Making”.

7. **The authors make the claim that empathy is a limited resource (L.60-2) but do not support this claim until L.79-86 (citations 22-6); I believe these lines should follow L.62. Additionally, the authors may wish to consult the following citations to further bolster their claim:**

- Bloom, P. (2017). *Against empathy: The case for rational compassion*. Random House.
- Cameron, C. D., & Payne, B. K. (2011). Escaping affect: how motivated emotion regulation creates insensitivity to mass suffering. *Journal of personality and social psychology*, 100(1), 1.
- Västfjäll, D., Slovic, P., Mayorga, M., & Peters, E. (2014). Compassion fade: Affect and charity are greatest for a single child in need. *PloS one*, 9(6), e100115.

We thank the reviewer for the suggestion to support our claim about empathy as a limited resource immediately after it is stated and cite additional supporting studies. Following the reviewer’s comment, we now cite additional support for the claim that empathy is limited (lines 86-87).

8. **The authors emphasize performance art in the Discussion (L.549-61), though I would encourage them to discuss it at greater length (earlier, and throughout) because it is the most novel component of their investigation and it is noteworthy and very interesting.**

We thank the reviewer for the positive feedback. In the revised manuscript, we added information about performance art and how we integrated it in the psychological experiments (Introduction: lines 146-160, Results: lines 377-398, Discussion: 714-724).

9. **This is entirely at the discretion of the authors, but I would encourage them to consider conducting a meta-analysis of the studies they present within this manuscript. I believe such an analysis would bolster their presentation of the robustness of their results, as well as providing a compelling figure to readers.**

Thank you for this wonderful suggestion! We ran a meta-analysis that included all four experimental studies (Studies 2-5). The results further strengthen our claims and are now reported in the manuscript (lines 560-569).

- 10. It would be helpful to readers to include the number of studies and the total sample size (by my count: 2,118; 1308+182+194+150+108+176) in the abstract (and perhaps also towards the end of the introduction, e.g. L.129).**

Following the reviewer's request, we added the number of studies and total sample size in the abstract (line 30) and at the end of the Introduction (line 136).

- 11. This paper has numerous grammatical, syntax, and miscellaneous language issues which need to be addressed. For example, L.41-2 is an incomplete sentence and the verb conjugation ("force") is incorrect:
"From the beginning of the 21st century, armed intergroup conflicts cost the lives of more than 1 million people worldwide (3), force more than 70 million people to flee their homes (4)."**

We thank the reviewer for this comment. We corrected all language issues throughout all the manuscript.

General comments on analyses and results

- 12. All figures should represent variance with error bars (currently missing from Figure 1). It is somewhat a question of style, though I think it is becoming more normative and I think it is more helpful to include 95% CI rather than SE.**

Following the reviewer's and the editor's comments we updated all bar charts to box-and-whisker plots that display centrality and dispersion measures.

- 13. I appreciate that the authors have included methodological details at the end of the manuscript (as per journal customs), though it would help readers to include basic details about their dependent measures throughout the manuscript (e.g. for Study 1, consider inserting the following text from Methods around L.180):
• L.654-57 "After reading the article, participants rated their empathic reactions (from 1 = not at all, to 7 = very much) toward the injured people ("Following the article, to what extent do you currently feel each of the feelings below? Empathy; Sympathy; Compassion"; $\alpha = .94$.)"**

Following the reviewer's request, we added a description of the dependent variables in the result section of each study (lines 196-198, 243-245, 293-296, 428-429, 505).

- 14. Although the authors detail their power calculations in the Methods, they should acknowledge that they computed these for each study early in the manuscript (and direct readers to the Methods for more detail).**

Following the reviewer's comment we added a note in the result section that power estimations can be found in the Method section (line 173).

- 15. In the Methods (e.g. Study 5), the authors provide separate Crombach's alphas for their multi-item dependent measures for each article. Perhaps I am ignorant of some methodological developments, but it is my impression that alphas should be calculated *across* condition, not within them. Therefore, one alpha value is sufficient (and should be presented in the main text, rather than the Methods, as indicated above).**

We apologize if our reliability calculations were unclear. The alphas are reported separately for each target (toward ingroup and toward outgroup) across all conditions.

- 16. This is a very minor style note on presenting condition means, but it would clarify the authors' writing to comprehensively check their manuscript to make the following edit:**
- **For example, L.446-448: "Such that participants in the unlimited condition perceived empathy as less limited compared to participants in the control condition (M = 5.01, SD = 1.38; M = 4.49, SD = 1.67 respectively)."**
 - **"Such that participants in the unlimited condition perceived empathy as less limited (M = 5.01, SD = 1.38) compared to participants in the control condition (M = 4.49, SD = 1.67)."**

Following the reviewer's comment, we changed the style of presenting condition means throughout the manuscript.

Pilot study

- 17. Table 1 documents (L.157) "A series of simple bivariate correlations (with no correction)" but it seems correction is warranted here because it presents multiple comparisons (i.e. empathic sentiments toward different groups with belief in empathy as a limited resource).**

Following the reviewer's suggestion, we conducted Bonferroni correction on the multiple correlations and reported it in the manuscript (lines 173-178).

- 18. Figure 1 is somewhat misleading; they artificially discretize "belief about empathy" (x-axis) yet plot lines which imply that points along the line are meaningful (which they are not in this context). Given that the authors are presenting the relationship between two continuous variables, why not superimpose regression lines (as they have done) over a scatterplot of the data?**

Following the reviewer's suggestion, we changed Figure 1 that now includes a scatterplot and the regression lines.

19. Study 1

♣ Figure 2 (as well as Figure 6 in Study 4)

- The data labels are confusing, in that it is unclear whether they index content or order (although they seem to imply the latter). In Figure S3, the authors present 4 testimonies (unlabeled) and write (L.231-2) that “all testimonies were presented in a counterbalanced order.” Therefore, does the label, e.g., “1st testimony” refer to “Alaa, 32, Daraya” or to the first testimony participants read?
- Also, the analysis the authors present is interesting, it is unclear how it bears on their hypotheses; why do you think participants became **more** empathic over the course of the “limited empathy” treatment? I would be inclined to keep this analysis—and to speculate on its implications—but I would not make this figure based on it.
- Instead, I would encourage the authors to test the interaction of testimony **content** and condition on empathic reactions. Presumably, the authors do not think that there is an effect of testimony content, and finding that this interaction is not significant (which would be my guess) would enable them to collapse the data across this dimension (making both figures more interpretable).

We thank the reviewer for the suggestion to present the interaction between condition (limited vs. unlimited) and the content of stories (four different testimonials) instead of the order of stories (1st, 2nd, 3rd, 4th) that was counterbalanced. Using the suggested analysis (with stories content) revealed no significant interaction. As such, we think that it would be better to focus on the order of the stories that allows us to examine potential changes in empathy over time.

Study 3

- 20. The authors present two separate analyses for their “empathic reactions” and “support for prosocial actions” dependent measures; yet, because these represent multiple observations from the same participants, a more appropriate analysis would employ multivariate regression.**

The reviewer suggested to run a multivariate regression that include both dependent variables (i.e., empathic reactions and support for prosocial actions). While it is possible to run such analysis, it could only add to the current analysis whether there is an interaction (or lack thereof) between the two dependent variables. Although we understand the merit of such analysis in other contexts, we believe that such interaction is less relevant in the current context and given that we had no initial hypotheses regarding such interaction. Thus, we prefer to keep the original analysis which we believe would be easier for readers to understand.

- 21. I appreciate the authors’ use of the Hayes PROCESS models, though I encourage them to ensure that they are implementing their mediation model correctly here, given that**

they employ a categorical independent variable (see citation below for guidance).
• Hayes, A. F., & Preacher, K. J. (2014). Statistical mediation analysis with a multicategorical independent variable. *British journal of mathematical and statistical psychology*, 67(3), 451-470.

We thank the reviewer for this helpful insight. Following the reviewer's recommendation, we re-ran the analysis with the condition defined as categorical and corrected it in the manuscript (lines 199-205).

Study 4

22. I believe there is a typo in the authors' reporting of means here:

• L.389-91 "On average, participants in the unlimited condition felt more empathy toward the refugees ($M = 5.5$, $SD = 1.25$), compared to participants in the limited condition ($M = 6.08$, $SD = 1.13$)."

Thank you for noticing this! Indeed, the values were switched accidentally, and we corrected it.

Reviewer #2

23. I've now read "Believing empathy is unlimited: Using performance art to reconstruct intergroup empathy and bring people together" with great interest. The work has several important strengths. First, it leverages a novel theoretical approach to intervening on empathy. Second, it offers a rigorous test of its hypotheses across several contexts, including multiple lab and field experiments. Third, it offers a cost-effective and scalable tool to address conflict. Finally, the manuscript is well written and easy to follow. In short, there was a lot to like here, and my overall evaluation was very positive.

Thank you for the positive feedback!

24. The authors assert that changing perceptions of empathy-related capacities could facilitate intergroup empathy. This is because some individuals consider empathy a limited resource, which they choose to save for ingroup members. I agree with the authors that this is a critically important area for inquiry. However, I wonder how the authors think about how beliefs about limitations of empathy—the focus of the present intervention—interact with other motivational forces that deter empathy for outgroup members.

For example, empathizing with outgroup members is seen by some as antithetical to the pursuit of group goals, especially in instances of (real or perceived) zero-sum competition. Empathizing with outgroup members may also be more difficult than empathizing with ingroup members due to perceived dissimilarities, and it could therefore be seen as more costly than empathizing with ingroup members [1,2]. It

would be informative to include some discussion on how beliefs about capacity interact with other empathic “avoidance motives”, such as potential interference with group goals, or the motivation to avoid costly empathizing. This could fit nicely with a related point the authors make about limitations of the intervention in the discussion (lines 583-588).

We agree with the reviewer that there are additional challenges in increasing empathy in conflictual and zero-sum competitions. Our research findings provide initial evidence that the belief about empathy as unlimited has an effect on empathic feelings and prosocial behavior, even in such contexts. However, we agree that in cases in which there are multiple conflicting motives, changing the belief that empathy is unlimited might not be sufficient. Accordingly, following the reviewer’s suggestion, in the revised manuscript, we extended the discussion and added a paragraph in the Discussion about such potential conflicting interactions and the importance of testing them (lines 684-697).

25. Along these lines, the manner in which the authors discuss “zero-sumness” related to empathy differs somewhat from how it is discussed by other researchers. Prior theoretical accounts explain how zero-sumness of resources may drive intergroup empathy differences, because individuals are motivated to amass resources for their ingroup. However, here the authors describe empathy itself as zero-sum, irrespective of resource limits (line 71). How do the authors imagine this paradigm to work in circumstance where resources remain zero-sum? Isn't it possible that people believe empathy is unlimited but still choose not to empathize with outgroup members? Why should changing this belief that empathy is unlimited help individuals overcome other empathic avoidance motives? This feels like an important theoretical point, and I would appreciate seeing this described in the introduction.

We thank the reviewer for this important comment. Indeed, empathy is not the only resource that people can consider in zero-sum terms as previous research pointed out. Moreover, previous studies found that people with zero-sum perception of a competition or resource held more negative attitudes toward outgroup members. As such, we believe that such zero-sum perceptions can either be aligned with or contradict the belief about empathy as an unlimited resource and its effect on empathic reactions and prosociality toward outgroup members. Following the reviewer’s suggestion, we added this theoretical point to the manuscript (lines 631-636 and 684-697).

26. Although group-based differences emerge clearly in every experiment, the absolute levels of empathy for outgroup members were higher than what I would have expected based on previous work (especially studies 2 and 4). I would appreciate it if the authors could offer comment on this, particularly because other studies find low levels of empathy for outgroup members (or even counter-empathy, such as schadenfreude) [3].

We agree with the reviewer that in a few studies the level of empathy was relatively higher than expected. One reason that might explain the relatively high levels of empathy toward outgroups is that participants were introduced with empathy-inducing stories that involve sad personal experiences (e.g., financial loss, health issue, torture) rather than neutral or collective sad stories. In addition, the intergroup relations between Syrian refugees and Americans are less conflictual compared to other contexts being examined, thus resulting in higher level of empathy. Following the reviewer's comment, we commented on this issue after reporting the results of Study 2 (lines 278-281).

27. I wonder whether the authors have any thoughts on the different aspects of the performance art intervention offering both first-person and third-person insights into others' experiences. Previous work finds that "imagine-self" and "imagine-other" perspective taking instructions have very different consequences [4]. From my read, the performance art intervention included experiences consistent with each of these types of perspective taking. It would be helpful to see a brief discussion addressing this aspect of their intervention.

The reviewer raised an interesting distinction between "imagine-self" versus "imagine-other" tactics for perspective-taking, which are known to have different consequences. Although we did not aim to examine the differences between these methods in the intervention, we considered this aspect when planning the performance art-experiments. It was important for us not to use "imagine-self" instructions that are known to increase personal distress (rather than empathic concern) and are negatively linked with avoiding prosocial behavior (Batson et al., 1997). Instead, the performers who shared the empathy-inducing stories focused on sharing their own experience without asking the participants to imagine themselves in their situation. Following the reviewer's suggestion, we now discuss this issue in Study 5 (lines 487-492).

C. D. Batson, S. Early, G. Salvarani, Perspective Taking: Imagining How Another Feels Versus Imaging How You Would Feel. *Pers. Soc. Psychol. Bull.* **23**, 751–758 (1997).

28. The authors describe collecting measures of sympathy and compassion for ingroup and outgroup members in addition to differences in empathy. Did the authors provide participants definitions of these terms, or otherwise distinguish empathy, sympathy and compassion? Were there group-based differences in sympathy and compassion ratings as well?

As the reviewer notes, in Studies 1,3, and 5 we averaged across empathy, sympathy, and compassion, because they all represent prosocial emotions and involve caring for others (e.g., Batson et al., 1997). Although empathy, sympathy and compassion differ from each other, our goal was to obtain a reliable index of prosocial emotions, and so we opted to use a more reliable, albeit conceptually broader measure. Internal reliability was above .7 in all studies,

supporting our assumption that the three terms overlap conceptually, at least to some extent. In the experimental studies, in which the belief about empathy was manipulated, participants received a definition of empathy. In other studies, we did not provide participants with definitions of any of the emotions that were measured. When examining effects of specific items, group-based differences were found in ratings of sympathy and compassion, similarly to empathy.

29. Were there any data exclusions in any of these experiments? If not, please state that explicitly.

Data exclusions are explicitly mentioned in lines 750-751, 783-784, 807-808.

In sum, this was a huge undertaking on the part of the researchers. I believe this article is of great theoretical and practical importance. It is an excellent piece of scholarship and with some revisions I hope to see it published at Nature Communications.

Thank you very much!

References

1. Zaki, J. (2014) Empathy: A motivated account. Psychol. Bull. 140, 1608–1647. <https://doi.org/10.1037/a0037679>
2. Zaki, J. and Cikara, M. (2015) Addressing Empathic Failures. Curr. Dir. Psychol. Sci. 24, 471–476. <https://doi.org/10.1177/0963721415599978>
4. Cikara, M. et al. (2014) Their pain gives us pleasure: How intergroup dynamics shape empathic failures and counter-empathic responses. J. Exp. Soc. Psychol. 55, 110–125. <https://doi.org/10.1016/j.jesp.2014.06.007>
5. Lamm, C. et al. (2007) The neural substrate of human empathy: effects of perspective-taking and cognitive appraisal. J. Cogn. Neurosci. 19, 42–58. <https://doi.org/10.1162/jocn.2007.19.1.42>

Reviewer #3 (Remarks to the Author):

A series of correlational and experimental studies are reported showing that leading participants to believe that empathy is unlimited (compared to limited) increased outgroup empathy.

This is a great paper and I enjoyed reading it. The theorising is sound, the studies are cleverly designed and competently conducted, producing noteworthy results. I particularly like the range of different groups and different conflicts that were studied including

different ethnic groups (Jews vs. Arabs, Syrian refugees). Most impressive was the final study including the manipulation through performance and the study of behaviours. I would love to see more studies like this.

I can't think of ways to improve this paper and think it is an important paper that holds great promise to move the field of reducing intergroup conflict forward in important ways. The limited versus unlimited empathy concept is novel and has great potential practically.

I soon hope to see this paper in print in Nature Communications.

Thank you for the positive feedback!

To conclude, in our revised manuscript and in this correspondence, we addressed all of the issues raised in your letter. We have done our best to address your and the reviewers' concerns, and we thank you for giving us the opportunity to clarify our positions on these issues. We hope that our answers are satisfying, and the revised manuscript is suitable for publication in Nature Communications.

REVIEWER COMMENTS

Reviewer #1 (Remarks to the Author):

The authors have completed a comprehensive revision of their original submission in light of my (and other reviewers') comments. Their response letter thoroughly detailed their edits, which they diligently endeavored in response to each reviewer criticism. Specifically, the authors have better situated their work within the literature, clarified their operationalization of their central construct (empathy), overhauled their figures, and adjusted their analyses to more appropriately (and thoroughly) test their research questions in accordance with their experimental design. Most importantly (in my estimation), they have provided a more detailed account of their novel performance art intervention, which I sincerely hope will inspire much future research! I am honored to have had the pleasure of serving as reviewer on this fantastic manuscript, and very much look forward to seeing it in print.

Reviewer #2 (Remarks to the Author):

The research team did a nice job addressing the comments. I was particularly delighted by the changes to the discussion. The work is thoughtful and rigorous-- well done.

The only thing I would still like to see as a reader is an explanation of the decision points laid out in response to comment 28 within the main text of the article. I don't feel strongly that the data be analyzed one way or another, and their explanation is reasonable to me. However, I do feel that this conceptual point would help inform an ongoing debate about the separability and overlap of distinct constructs related to empathy (e.g., Weisz E, Cikara M. Strategic Regulation of Empathy. *Trends Cogn Sci.* 2021 Mar;25(3):213-227. doi: 10.1016/j.tics.2020.12.002. Epub 2020 Dec 30. PMID: 33386247).

I would encourage the authors to explore their decision to collapse constructs in the main text, and (even better) discuss what happens when they're analyzed separately in order to help advance our understanding of when these constructs separate and when they do not. However, I leave it to the discretion of the editor to decide whether or not this modification should be implemented before publication.

Congratulations to the research team on an excellent piece of scholarship!

March 11, 2022

On behalf of the authors, I would like to thank the two reviewers for your careful reading and constructive review of our manuscript, titled "Using performance art to promote intergroup prosociality by cultivating the belief that empathy is unlimited" (# NCOMMS-21- 34614A). In the revised manuscript, we addressed all comments raised by the reviewers. We believe that the revised version is stronger and clearer than the previous submission, and we thank you for that. We hope you find the new version suitable for publication in Nature Communications. Below you can find our reply for each of the comments:

Reviewer #1

- 1. The authors have completed a comprehensive revision of their original submission in light of my (and other reviewers') comments. Their response letter thoroughly detailed their edits, which they diligently endeavored in response to each reviewer criticism. Specifically, the authors have better situated their work within the literature, clarified their operationalization of their central construct (empathy), overhauled their figures, and adjusted their analyses to more appropriately (and thoroughly) test their research questions in accordance with their experimental design. Most importantly (in my estimation), they have provided a more detailed account of their novel performance art intervention, which I sincerely hope will inspire much future research! I am honored to have had the pleasure of serving as reviewer on this fantastic manuscript, and very much look forward to seeing it in print.**

Thank you for the positive feedback!

Reviewer #2

- 2. The research team did a nice job addressing the comments. I was particularly delighted by the changes to the discussion. The work is thoughtful and rigorous-- well done.**

Thank you for the positive feedback!

- 3. The only thing I would still like to see as a reader is an explanation of the decision points laid out in response to comment 28 within the main text of the article. I don't feel strongly that the data be analyzed one way or another, and their explanation is reasonable to me. However, I do feel that this conceptual point would help inform an ongoing debate about the separability and overlap of distinct constructs related to empathy (e.g., Weisz E, Cikara M. Strategic Regulation of Empathy. Trends Cogn Sci. 2021 Mar;25(3):213-227. doi: 10.1016/j.tics.2020.12.002. Epub 2020 Dec 30. PMID:**

33386247).

I would encourage the authors to explore their decision to collapse constructs in the main text, and (even better) discuss what happens when they're analyzed separately in order to help advance our understanding of when these constructs separate and when they do not. However, I leave it to the discretion of the editor to decide whether or not this modification should be implemented before publication.

Congratulations to the research team on an excellent piece of scholarship!

Following the reviewer's suggestion, we added an explanation in the manuscript for using measures of empathy, sympathy and compassion (lines 201-205) and mentioned that although they are related to each other and represent prosocial emotions, they still have distinct properties that can lead to different emotional and behavioral outcomes.

Moreover, following the reviewer's request to run the analyses for each emotion separately, we found similar trends such that among participants in the control or limited conditions there were significant difference between ingroup vs. outgroup empathy, sympathy, and compassion, while such differences were not significant among participants in the unlimited condition.